# Understanding the Generalization Benefit of Model Invariance from a Data Perspective

**Sicheng Zhu\*, Bang An\*, Furong Huang**
Department of Computer Science
University of Maryland, College Park
{sczhu, bangan, furongh}@umd.edu

## Abstract

Machine learning models that are developed to be invariant under certain types of data transformations have shown improved generalization in practice. However, a principled understanding of why invariance benefits generalization is limited. Given a dataset, there is often no principled way to select "suitable" data transformations under which model invariance guarantees better generalization. This paper studies the generalization benefit of model invariance by introducing the *sample cover induced by transformations*, i.e., a representative subset of a dataset that can approximately recover the whole dataset using transformations. For any data transformations, we provide refined generalization bounds for invariant models based on the sample cover. We also characterize the "suitability" of a set of data transformations by the *sample covering number induced by transformations*, i.e., the smallest size of its induced sample covers. We show that we may tighten the generalization bounds for "suitable" transformations that have a small sample covering number. In addition, our proposed sample covering number can be empirically evaluated and thus provides a guidance for selecting transformations to develop model invariance for better generalization. In experiments on multiple datasets, we evaluate sample covering numbers for some commonly used transformations and show that the smaller sample covering number for a set of transformations (e.g., the 3D-view transformation) indicates a smaller gap between the test and training error for invariant models, which verifies our propositions.

## 1 Introduction

Invariance is ubiquitous in many real-world problems. For instance, categorical classification of visual objects is invariant to slight viewpoint changes [18, 2, 23], text understanding is invariant to synonymous substitution and minor typos [53, 36, 27]. Intuitively, models capturing the underlying invariance exhibit improved generalization in practice [21, 13, 50, 14, 45, 14]. Such generalization benefit is especially crucial when the data are scarce as in some medical tasks [46], or when the task requires more data than usual as in cases of distribution shift [38] and adversarial attack [40, 49, 5].

A commonly accepted intuition attributes the generalization benefit of model invariance to the reduced model complexity, especially the reduced sensitivity to spurious features. However, a principled understanding of why model invariance helps generalization remains elusive, thus leaving many open questions. Since model invariance may come at a cost (e.g., compromised accuracy, increase computational overhead), given a task, how should we choose among various data transformations under which model invariance guarantees better generalization? If existing data transformations are not good enough for a given task, what is the guiding principle to find new ones? The lack of a principled understanding limits better exploitation of model invariance to further improve

---

\* Equal contribution.

35th Conference on Neural Information Processing Systems (NeurIPS 2021).

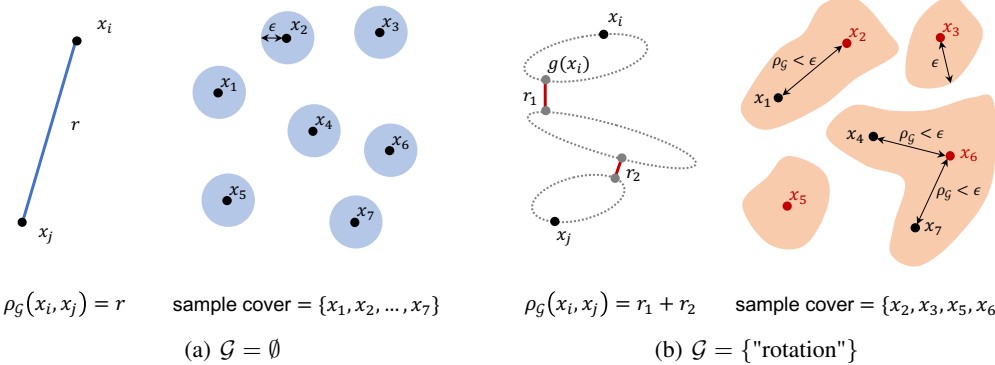

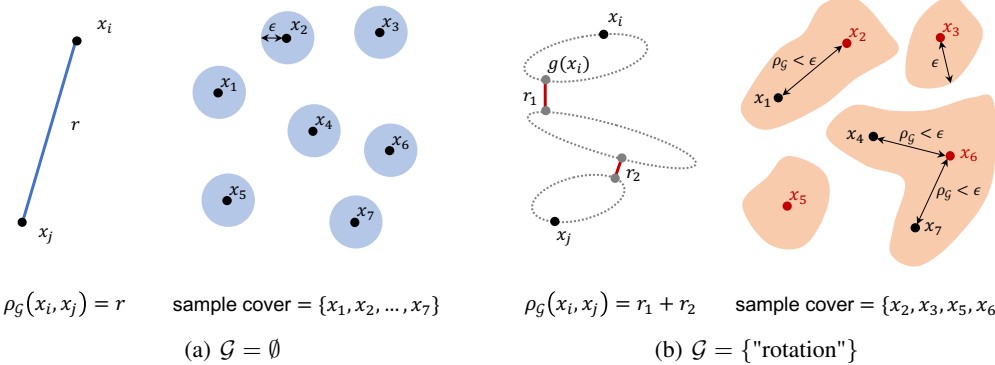
$$\rho_{\mathcal{G}}(x_i, x_j) = r \qquad \text{sample cover} = \{x_1, x_2, \dots, x_7\} \qquad\qquad \rho_{\mathcal{G}}(x_i, x_j) = r_1 + r_2 \qquad \text{sample cover} = \{x_2, x_3, x_5, x_6\}$$

(a) $\mathcal{G} = \emptyset$            (b) $\mathcal{G} = \{\text{"rotation"}\}$

Figure 1: Illustration of the pseudometric and sample cover induced by data transformations.

generalization. In addition, since identifying instructive generalization bound is a central topic in machine learning, we may expect to tighten existing generalization bounds by additionally considering the data-dependent model invariance property.

The many faces of data transformations and model classes pose significant challenges to a principled understanding of model invariance's generalization benefit. To address this, [2, 4, 34, 42, 39] characterize the input space and show that certain data transformations equivalently shrink the input space for invariant models, which then simplify the input and improves generalization. From another perspective, [19, 30] directly characterize the function space and show that the volume of the invariant model class is reduced, which then simplifies the learning problem and improves generalization. These understandings provide valuable insights, yet they may become less informative on high-dimensional input data or require the model invariance to be obtained exclusively via feature averaging. Some certain assumptions on data transformations (e.g., finiteness, group structure with certain measure) also make these understandings less applicable to more general data transformations.

In this paper, we derive generalization bounds for invariant models based on the sample cover induced by data transformations and empirically show that the introduced notion can guide the data transformations selection. Different from previous understandings, we first identify a data-dependent property of data transformations in a model-agnostic way, and then establish its connections with the refined generalization bounds of invariant models. The analysis applies to more general data transformations regardless of how the model invariance is obtained and naturally provides a model-agnostic guidance for data transformations selection. We summarize our contributions as follows.

**Contributions.** At the core of our understanding is the notion of sample cover induced by data transformations, defined informally as a representative subset of a dataset that can approximately recover the whole dataset using data transformations (illustrated in Figure 1). We show that this notion identifies a data-dependent property of data transformations which is related to the generalization benefit of the corresponding invariant models. Under a special setting of the sample cover, we first bound the model complexity of any invariant and output-bounded model class in terms of the sample covering numbers. Since this general bound requires a restrictive condition on data transformations in order to be informative, we then assume the model Lipschitzness to relax the requirement and refine the model complexity bound for invariant models. Finally, we outline a framework for model-invariance-sensitive generalization bounds based on the invariant models' complexities, and use it to discuss the generalization benefit of model invariance.

Given the usefulness of sample cover in the analysis, we propose an algorithm to empirically estimate the sample cover. This algorithm exactly verifies whether a given subset of a sample forms a valid sample cover, and always estimates a sample covering number that upper-bounds the ground-truth. Inspired by our analysis, we also propose to use the sample covering number as a suitability measurement for practical data transformation selections. This measurement is data-driven, widely applicable, and empirically correlates with invariant models' actual generalization performance. We discuss its limitations and the empirical mitigation.

To empirically verify our propositions, we first estimate the sample covering number for some commonly used data transformations on four image datasets, including CIFAR-10 and ShapeNet (a 3D dataset). Under typical settings, the 3D-view transformation induces a much smaller sample covering number than others on ShapeNet, while cropping induces the smallest sample covering number on others datasets. For those data transformations, we then train invariant models via data augmentation and invariance loss regularization to evaluate the actual generalization benefit. Results show a clear correlation between smaller sample covering numbers induced by data transformations and the better generalization benefit enjoyed by invariant models.

## 2 Preliminaries

**Data transformations.** We refer to the data transformation as a function from the input space $\mathcal{X} \to \mathcal{X}$, and data transformation*s* as a set of such functions. Unless otherwise specified, we do not assume data transformations to have group structures since many non-invertible transformations (e.g., cropping) do not fit into a group structure directly. For a set of data transformations $\mathcal{G} = \{g : \mathcal{X} \to \mathcal{X}\}$ and a data point (also referred to as an example) $\boldsymbol{x} \in \mathcal{X}$, we overload the notion of orbit in group theory and denote by $\mathcal{G}(\boldsymbol{x})$ the orbit of $\boldsymbol{x}$ defined as follows. The *orbit* of $\boldsymbol{x}$ generated by data transformations $\mathcal{G}$ is the collection of outputs after applying any transformation $g \in \mathcal{G}$ on $\boldsymbol{x}$: $\mathcal{G}(\boldsymbol{x}) = \{g(\boldsymbol{x}) \in \mathcal{X} : g \in \mathcal{G}\}$.

**Model invariance.** Let $\mathcal{D}$ be the underlying data distribution and supp$(\mathcal{D})$ be its support. A model $h : \mathcal{X} \to \mathcal{Y}$ is said to be *invariant* under data transformations $\mathcal{G}$ on $\mathcal{D}$ if $h(g(\boldsymbol{x})) = h(\boldsymbol{x})$ for any $\boldsymbol{x} \in$ supp$(\mathcal{D})$ and any $g \in \mathcal{G}$. We refer to a class of invariant models as the $\mathcal{G}$-*invariant model class*.

**Complexity measurements.** *Covering number* and *Rademacher complexity* [33] are two commonly used complexity measurements for model classes (including neural networks [6]) that can provide uniform generalization bounds. The covering number can also be directly used to upper bound the Rademacher complexity via Dudley's entropy integral theorem [17, 32].

*Covering number.* Let $(\mathcal{F}, d)$ be a (pseudo)metric space with some (pseudo)metric[1] $d$. An $\epsilon$-*cover* of a set $\mathcal{H} \subseteq \mathcal{F}$ is defined as a subset $\widehat{\mathcal{H}} \subseteq \mathcal{H}$ such that for any $h \in \mathcal{H}$, there exists $\widehat{h} \in \widehat{\mathcal{H}}$ such that $d(h, \widehat{h}) \leq \epsilon$. The covering number $N(\epsilon, \mathcal{H}, d)$ is defined as the minimum cardinality of an $\epsilon$-cover (among all $\epsilon$-covers) of $\mathcal{H}$. In this paper, we use the concept of covering number both for measuring model class complexities and for defining the sample covering number on datasets.

*Empirical Rademacher complexity.* Let $\mathcal{H}$ be a class of functions $h : \mathcal{X} \to \mathbb{R}$. Given a sample $\mathcal{S} = \{\boldsymbol{x}_i\}_{i=1}^n$, the *empirical Rademacher complexity* of model class $\mathcal{H}$ is defined as: $\mathfrak{R}_{\mathcal{S}}(\mathcal{H}) = \mathbb{E}_{\boldsymbol{\sigma}} \left[ \sup_{h \in \mathcal{H}} \frac{1}{n} \sum_{i=1}^n \sigma_i h(\boldsymbol{x}_i) \right]$ where $\boldsymbol{\sigma} = [\sigma_1, ..., \sigma_n]^\top$ is the vector of i.i.d. Rademacher random variables, each uniformly chosen from $\{-1, 1\}$.

**Generalization error and gap.** Let $\mathcal{S} = \{\boldsymbol{x}_i\}_{i=1}^n$ be a sample drawn i.i.d. from some data distribution $\mathcal{D}$, and $\mathcal{H}$ be a model class. Given a loss function $\ell : \mathbb{R} \to [0, 1]$, for a $h \in \mathcal{H}$, we define the *empirical error* as $R_{\mathcal{S}}(h) = \frac{1}{n} \sum_{i=1}^n \ell(h(\boldsymbol{x}_i), y_i)$, the *generalization error* as $R(h) = \mathbb{E}_{(\boldsymbol{x}, y) \sim \mathcal{D}}[\ell(h(\boldsymbol{x}), y)]$, and the *generalization gap* as $R(h) - R_{\mathcal{S}}(h)$.

## 3 Generalization Benefit of Model Invariance

In this section, we derive the generalization bounds for invariant models by identifying the model invariance properties. We start by introducing the notion of sample cover induced by data transformations and based on it bound the Rademacher complexity of any invariant models with bounded output (Section 3.1). Then, we assume model Lipschitzness to provide a more informative model complexity bound for any data transformations (Section 3.2). Finally, we provide a framework for model-invariance-sensitive generalization bounds and discuss the generalization benefit of model invariance (Section 3.3).

---

[1] A pseudometric is a metric if and only if it separates distinct points, namely $d(x, y) > 0$ for any $x \neq y$.

## 3.1 Sample Cover Induced by Data Transformations

Existing empirical results suggest that, compared with standard models, invariant models may have certain properties reducing their effective model complexities. To identify such properties, we alternatively identify the related properties of the corresponding data transformations via the notion of *sample cover induced by data transformations*. We now formalize the introduced notion.

The definition of sample cover relies on the pseudometric induced by the data transformations $\mathcal{G}$. Note that $\mathcal{G}$ generates an orbit $\mathcal{G}(\boldsymbol{x}) \subseteq \mathcal{X}$ for each example $\boldsymbol{x} \in \mathcal{S}$. Let $\|\cdot\|$ be any norm on the input space $\mathcal{X}$. Given a set of transformations $\mathcal{G}$, we define the $\mathcal{G}$-induced pseudometric[2] as

$$\rho_{\mathcal{G}}(\boldsymbol{x}_1, \boldsymbol{x}_2) = \inf_{\gamma \in \Gamma(\boldsymbol{x}_1, \boldsymbol{x}_2)} \int_{\gamma} c(\boldsymbol{r}) ds, \quad \text{where } c(\boldsymbol{r}) = \begin{cases} 0, & \text{if } \boldsymbol{r} \in \cup_{\boldsymbol{x} \in \mathcal{S}} \mathcal{G}(\boldsymbol{x}) \\ 1, & \text{otherwise} \end{cases} \tag{3.1}$$

where $ds = \|d\boldsymbol{r}\|$, and $\Gamma$ denotes the set of all paths (curves) in $\mathcal{X}$ from $\boldsymbol{x}_1$ to $\boldsymbol{x}_2$. The $\rho_{\mathcal{G}}$ is essentially calculating the line integral along the shortest (if achievable) path $\gamma$ in the scalar field $c$, where $c$ can also be viewed as the "moving cost" function depending on $\mathcal{G}$. The norm $\|\cdot\|$ here can be selected as any meaningful norm on the input space (e.g., Euclidean norm as in our experiments) and will later be used in defining the model's Lipschitz constant. It can be checked that $\rho_{\mathcal{G}}$ satisfies pseudometric axioms.

**Definition 3.1** (Sample cover induced by data transformations). Let $(\mathcal{X}, \rho_{\mathcal{G}})$ be a pseudometric space and $\mathcal{S} = \{\boldsymbol{x}_i\}_{i=1}^n$ be a sample of size $n$. An $\epsilon$-*sample cover* $\widehat{\mathcal{S}}_{\mathcal{G},\epsilon}$ of the sample $\mathcal{S}$ induced by data transformations $\mathcal{G}$ at resolution $\epsilon$ is defined as a subset of the sample $\mathcal{S}$ such that for any $\boldsymbol{x} \in \mathcal{S}$, there exists $\widehat{\boldsymbol{x}} \in \widehat{\mathcal{S}}_{\mathcal{G},\epsilon}$ satisfying $\rho_{\mathcal{G}}(\boldsymbol{x}, \widehat{\boldsymbol{x}}) \leq \epsilon$.

**Definition 3.2** (Sample covering number induced by data transformations). The *sample covering number* $N(\epsilon, \mathcal{S}, \rho_{\mathcal{G}})$ induced by data transformations $\mathcal{G}$ is defined as the minimum cardinality of an $\epsilon$-sample cover:

$$N(\epsilon, \mathcal{S}, \rho_{\mathcal{G}}) = \min\{|\widehat{\mathcal{S}}_{\mathcal{G},\epsilon}| : \widehat{\mathcal{S}}_{\mathcal{G},\epsilon} \text{ is an } \epsilon\text{-sample cover of } \mathcal{S}\}. \tag{3.2}$$

Informally, the $\mathcal{G}$-induced sample cover specifies a representative subset of examples which can approximately recover all the original examples using the given data transformations $\mathcal{G}$. This notion is closely related to the *sample compression* [20] which represents a scheme to prove the learnability of concepts through a compressed set of sample. While identifying the generalization-related properties of data transformations, this notion is insensitive to other unrelated properties (e.g., finiteness, group structures) and thus applies to any data transformations.

The intuition behind sample cover is that $\mathcal{G}$-invariant models may have consistent behaviors on an example and its associated approximation in the $\mathcal{G}$-induced sample cover. As such, we can analyze the model complexities of invariant models by considering the models' behaviour only on the potentially small-sized sample covers. Indeed, we directly have the following model complexity result. The proof is in Appendix B.

**Proposition 3.3.** Let $\mathcal{S} = \{\boldsymbol{x}_i\}_{i=1}^n$ be a sample of size $n$. Let $\mathcal{H}$ be a model class mapping from $\mathcal{X}$ to $[-B, B]$ for some $B > 0$ and is invariant to data transformations $\mathcal{G}$. Then the empirical Rademacher complexity of $\mathcal{H}$ satisfy

$$\mathfrak{R}_{\mathcal{S}}(\mathcal{H}) \leq 24B \sqrt{\frac{N(0, \mathcal{S}, \rho_{\mathcal{G}})}{n}}. \tag{3.3}$$

Proposition 3.3 generally bounds the model complexity of any output-bounded and $\mathcal{G}$-invariant model class in terms of the sample covering number $N(0, \mathcal{S}, \rho_{\mathcal{G}})$ induced by $\mathcal{G}$. A small $\mathcal{G}$-induced sample covering number at resolution $\epsilon = 0$ thus tightens the model complexity bound for a general class of $\mathcal{G}$-invariant models.

Note, however, that Proposition 3.3 is informative only when the data transformations $\mathcal{G}$ yields $N(0, \mathcal{S}, \rho_{\mathcal{G}}) \ll n$ on the sample $\mathcal{S}$ — a condition requiring $\mathcal{G}$ to be able to exactly recover $\mathcal{S}$ from a small-sized subset of $\mathcal{S}$. This condition is unfortunately too strict to hold for many commonly used

---

[2]Note that $\rho_{\mathcal{G}}$ is not a metric since it allows $\rho_{\mathcal{G}}(x, y) = 0$ for $x \neq y$.

data transformations which only generate orbits with measure zero (with respect to the data measure) at most data points. For example, the rotation transformations on CIFAR-10 do not satisfy this condition, since no two images in CIFAR-10 are rotated versions of each other. To better understand the generalization benefit brought by any data transformations (e.g., rotation), we further assume specific model properties which equivalently expand the orbits in order to get more general results. We study Lipschitz models in Section 3.2, and relegate a sharper (and relatively independent) analysis for linear models under linear data transformations to Appendix C.

## 3.2 Refined Complexity Analysis of Lipschitz Models

This subsection refines the model complexity analysis for Lipschitz models that are invariant. Characterizing the Lipschitz constant of models has been the focus of a line of work. For example, the Lipschitz constant of ReLU networks can be upper-bounded by the product of the spectral norms of the weight matrices, considering the worst-case inputs [6, 22]. Assuming Lipschitzness, the following theorem refines the covering number analysis for invariant models. The proof is in Appendix B.

**Theorem 3.4.** Let $\mathcal{S} = \{\boldsymbol{x}_i\}_{i=1}^n$ be a sample of size $n$. Let $\mathcal{H}$ be a model class such that every $h \in \mathcal{H}$ is $\kappa$-Lipschitz with respect to $\|\cdot\|$ (used in defining the sample cover) and is invariant to $\mathcal{G}$. Then the covering number of $\mathcal{H}$ satisfies

$$N\big(\tau, \mathcal{H}, L_2(\mathbb{P}_\mathcal{S})\big) \leq \inf_{\epsilon \geq 0, \widehat{\mathcal{S}}_{\mathcal{G},\epsilon}} N\big(\tau - \kappa\epsilon\sqrt{1 - \frac{|\widehat{\mathcal{S}}_{\mathcal{G},\epsilon}|}{n}}, \mathcal{H}, L_2(\mathbb{P}_{\widehat{\mathcal{S}}_{\mathcal{G},\epsilon}})\big), \qquad (3.4)$$

where $\forall h, g \in \mathcal{H}$, the $L_2(\mathbb{P}_\mathcal{S})$ metric is defined as $\|h - g\|_{L_2(\mathbb{P}_\mathcal{S})} = \left(\sum_{\boldsymbol{x} \in \mathcal{S}} \frac{1}{n}\big(h(\boldsymbol{x}) - g(\boldsymbol{x})\big)^2\right)^{\frac{1}{2}}$, and the $L_2(\mathbb{P}_{\widehat{\mathcal{S}}_{\mathcal{G},\epsilon}})$ metric is defined as[3] $\|h - g\|_{L_2(\mathbb{P}_{\widehat{\mathcal{S}}_{\mathcal{G},\epsilon}})} = \left(\sum_{\boldsymbol{x} \in \widehat{\mathcal{S}}_{\mathcal{G},\epsilon}} \frac{p(\boldsymbol{x})}{n}\big(h(\boldsymbol{x}) - g(\boldsymbol{x})\big)^2\right)^{\frac{1}{2}}$.

Theorem 3.4 upper-bounds the covering number of $\mathcal{H}$ evaluated at the sample $\mathcal{S}$ by the new covering number evaluated at any sample cover $\widehat{\mathcal{S}}_{\mathcal{G},\epsilon}$, under a modified metric and at the cost of an additional error term depending on $\epsilon$ and $\kappa$. The equality trivially holds by taking $\widehat{\mathcal{S}}_{\mathcal{G},\epsilon} = \mathcal{S}$, while by searching over all sample covers with different resolution $\epsilon$ it is possible to tighten the covering number bound for invariant models. Additionally, Theorem 3.4 leads to a refined version of Dudley's entropy integral theorem (see Lemma B.1) that bounds the Rademacher complexity of invariant models.

Theorem 3.4 suggests that we may improve existing covering-number-based model complexity analysis by weakening the dependence on input dimensions. Note that covering numbers that do not yield $N\big(\tau, \mathcal{H}, L_2(\mathbb{P}_\mathcal{S})\big)/n \to 0$ as $n \to \infty$ are vacuous. Therefore, existing covering number results typically avoid linear dependence on $n$ at the cost of (explicitly or implicitly) increased dependence on the input dimension [52]. With the refined result in Theorem 3.4, however, a covering number linear in $n$ can now be replaced by one that is linear in a potentially much smaller sample covering number $N(\epsilon, \mathcal{S}, \rho_\mathcal{G})$ and consequently become informative, thus circumvent the increased dependence on input dimensions. An interesting direction for future work is to instantiate the result in Equation 3.4 for specific model classes to get more interpretable results.

## 3.3 Framework for Model-invariance-sensitive Generalization Bounds

This subsection presents the framework for generalization bounds sensitive to the model invariance. While the results are straightforward applications of the derived complexities of invariant models, our goal is to justify the selection of suitable data transformations to maximize the generalization benefit. We start with the generalization analysis of invariant models and then present the framework.

**Generalization benefit for invariant models.** The generalization bounds of invariant models follow immediately by applying the Rademacher model complexities (Proposition 3.3, Proposition B.1, and Theorem C.1) to the standard generalization bound (Theorem A.2). Compared with standard models, invariant models' tightened model complexity bounds already imply their reduced generalization gaps,

---

[3]The term $p(\boldsymbol{x})/n$ can be viewed as the probability mass at $\boldsymbol{x}$ where the numerator indicates the number of examples that $\boldsymbol{x}$ covers. See Appendix B.1 for the formal definition of $p(\boldsymbol{x})$.

whereas for reduced generalization error they further need to have low empirical error. Since enforcing the model invariance may simultaneously increase the empirical error, we can use standard model selection techniques (e.g., structural risk minimization [33]) to select suitable data transformations and control the trade-off.

**Model-invariance-sensitive generalization bound.** We outline the generalization bound that identifies model invariance properties based on the derived invariant models' complexities. It follows by the post-hoc analysis which specifies a proper set of invariant models using the "invariant loss" — the loss when composed with any model, makes the composition invariant. For data transformations with group structures, we can construct such loss by averaging (assuming Haar measure) or adversarially perturbing any given loss over the orbits of input examples [30, 19]. Specifically, the adversarial loss with respect to data transformations $\mathcal{G}$ is defined as $\widetilde{\ell}_{\mathcal{G}}(h(x), y) = \max_{x' \in \mathcal{G}(x)} \ell(h(x'), y)$, where $\ell$ is any given loss. Using the adversarial loss, the following proposition provides the model-invariance-dependent generalization bound by applying the model selection framework [33]. Appendix B.3 further describes a binary coding construction of combinations of data transformation classes.

**Proposition 3.5.** Let $\mathcal{S} = \{x_i\}_{i=1}^n$ be a sample of size $n$. Let $\mathcal{H}$ be any given model class and $\ell$ be any given loss. Suppose we have $K$ sets of group-structured data transformations $\{\mathcal{G}_1, \mathcal{G}_2, ..., \mathcal{G}_K\}$. Then with probability at least $1 - \delta$, the following generalization bound holds for any $h \in \mathcal{H}$ and any $k \in [K]$:

$$R(h) \leq \frac{1}{n} \sum_{i=1}^n \widetilde{\ell}_{\mathcal{G}_k}(h(x_i), y_i) + 4\mathfrak{R}_{\mathcal{S}}(\widetilde{\ell}_{\mathcal{G}_k} \circ \mathcal{H}) + \sqrt{\frac{\log k}{n}} + 3\sqrt{\frac{\log \frac{4}{\delta}}{2n}}, \qquad (3.5)$$

where $\mathfrak{R}_{\mathcal{S}}(\widetilde{\ell}_{\mathcal{G}_k} \circ \mathcal{H})$ is upper-bounded by the complexity of $\mathcal{G}_k$-invariant models. For any model trained on $\mathcal{S}$, Proposition 3.5 shows that we can optimize over all selections of data transformations to improve its generalization bound. Note that the selection of $\mathcal{G}_k$ is subject to a potential trade-off between the reduced model complexity $\mathfrak{R}_{\mathcal{S}}(\widetilde{\ell}_{\mathcal{G}_k} \circ \mathcal{H})$ and the increased empirical error $\sum_{i=1}^n \widetilde{\ell}_{\mathcal{G}_k}(h(x_i), y_i)$. Thus, if a suitable $\mathcal{G}_k$ reduces the model complexity while keeping the empirical error low, then the trained model will benefit from a tightened generalization bound. This generalization bound does not require the models to be (strictly) invariant and potentially explains the improved generalization of models with trained invariance (e.g., via data augmentation [43, 41] or consistency regularization [31, 47]). The difficulty in instantiating Proposition 3.5 is that the model complexity with adversarial loss may be hard to compute for general data transformations. Therefore, we discuss more practical data transformations selections based on the sample covering numbers in Section 5.

## 4  Sample Cover Estimation Algorithm

The sample cover induced by data transformations plays a central role in our understanding of model invariance. Despite the usefulness in the analysis, exactly computing the sample cover turns out to be non-trivial in general. Indeed, computing the transformation-induced metrics can be difficult for continuous data transformations, and finding the *smallest* sample cover is NP-hard. To address this problem, we propose an algorithm to estimate the sample covering number and find the associated sample cover. We outline the algorithm and discuss the algorithmic challenges in this section. The algorithmic details appear in Appendix D.

**Setup.** The estimation algorithm takes as input a sample $\mathcal{S}$, a set of data transformations $\mathcal{G}$, and the resolution parameter $\epsilon$. It then returns the estimated sample covering number $N(\epsilon, \mathcal{S}, \rho_{\mathcal{G}})$ and the associated sample cover $\widehat{\mathcal{S}}_{\mathcal{G}, \epsilon}$. The estimation algorithm has the following steps.

**Step 1.** Compute (or approximate) the direct orbit distance between any two examples in $\mathcal{S}$. The direct orbit distance between any two examples $x_i, x_j \in \mathcal{S}$ is

$$d_{\mathcal{G}}(x_i, x_j) = \|\mathcal{G}(x_i) - \mathcal{G}(x_j)\| = \min_{g_1, g_2 \in \mathcal{G}} \|g_1(x_i) - g_2(x_j)\|,$$

which can be exactly computed for finite transformations (e.g., flipping) with complexity $O(|\mathcal{G}|^2)$, or can be approximated for continuous transformations (e.g., rotation) via optimization or sampling.

**Step 2.** Compute the $\rho_{\mathcal{G}}$ distance between any two examples in $\mathcal{S}$. Given results in step 1, computing the $\rho_{\mathcal{G}}$ distance between any two examples can be formulated as a shortest path problem on a complete graph, where each node represents an example and the cost of each edge is the direct orbit distance computed in step 1 (see formulations in Appendix D). Note that the shortest path is always included in our finite candidates even though the $\rho_{\mathcal{G}}$ distance considers infinitely many paths. This is because any other path outside our finite candidates will be longer than its counterparts (depending on what orbits it intersects) in our finite candidates. Standard shortest path algorithms solve for all pairs of examples in polynomial time (e.g., via Dijkstra's algorithm [16] in $O(n^3)$).

**Step 3.** Construct the pairwise distance matrix $[\rho_{\mathcal{G}}(\boldsymbol{x}_i, \boldsymbol{x}_j)]_{i,j}$ and approximate the sample covering number. This step can be formulated as a set cover problem where each example $\boldsymbol{x}$ covers a subset of $\mathcal{S}$ in which each element's $\rho_{\mathcal{G}}$ distance to $\boldsymbol{x}$ is less than or equal to $\epsilon$. Our goal is to find a minimum number of those subset such that their union contains $\mathcal{S}$. This problem is known to be NP-hard in general but admits polynomial time approximations [24]. In experiments, we use modified k-medoids [35] clustering method to find the approximation of $N(\epsilon, \mathcal{S}, \rho_{\mathcal{G}})$ (see Algorithm 1).

Note that the estimated sample covering number returned by the algorithm is always an upper bound of the ground-truth, regardless of the approximation error in step 1 and 3. When step 1 is exact, the algorithm also exactly verifies whether a given subset of $\mathcal{S}$ forms a valid sample cover. In our experiment, the step 2 becomes the computation bottleneck for large-sized sample. We leave improving the scalability as well as evaluating the approximation quality for future work.

## 5 Data-driven Selection of Data Transformations

The pool of candidate data transformations on a given dataset may be infinitely large. To maximize the generalization benefit of model invariance, we usually make selections based on expensive cross-validations due to the absence of a model-training-free guidance. Section 3 suggests that invariant models may benefit from improved generalization guarantees if the corresponding data transformations induce small sample covering numbers. Therefore, we propose to use the sample covering number as an empirical suitability measurement to guide the data transformations selection. We discuss its advantages, limitations, and empirical mitigation in this section.

**Suitability measurement.** To maximize the generalization benefit of model invariance on a dataset $\mathcal{S}$, we measure the suitability of data transformations $\mathcal{G}$ by the sample covering number induced by $\mathcal{G}$ and favor the small ones.

**Advantages.** One advantage of this suitability measurement is that it is model-training-free. It provides a-priori guidance depending only on the dataset and the data transformations, thus avoids expensive cross-validations and fuels the exploration of new types of data transformations. Another advantage is that it applies to any types of data transformations (including the continuous and non-invertible ones) and provides a uniform benchmark.

**Limitations and empirical mitigation.** Being model-agnostic also poses two limitations to the suitability measurement. One limitation is that this suitability measurement, while capturing invariant models' reduced generalization gap, ignores their potentially increased empirical error. Note that certain data transformations on a dataset may drastically increase invariant models' empirical error and overturn the benefit of reduced generalization gap. To mitigate this limitation, we consider two necessary conditions for maintaining low empirical error. First, the data transformations should preserve the underlying ground-truth labeling. We may use domain knowledge to meet this condition. Second, the model class should be rich enough to contain a low-error invariant hypothesis. In our experiment, neural networks which are invariant and achieve low training error suffice this condition.

Another limitation is that this suitability measurement ignores models' potential Lipschitz constant change after enforcing the invariance. Theorem 3.4 suggests that the generalization benefit enjoyed by invariant models depends on models' Lipschitz constant and can be overturned if enforcing invariance leads to significantly larger Lipschitz constant. To mitigate this limitation, we use the fact that we are doing classification tasks and use the label information to heuristically offset the Lipschitz constant increase. We use the minimum inter-class distance change after applying data transformations to

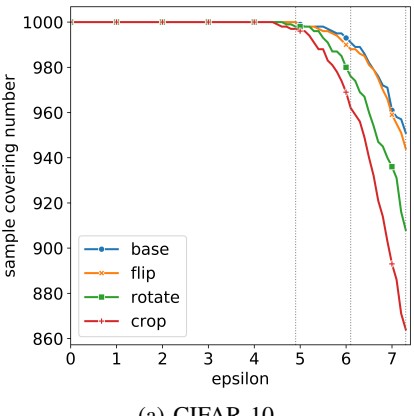 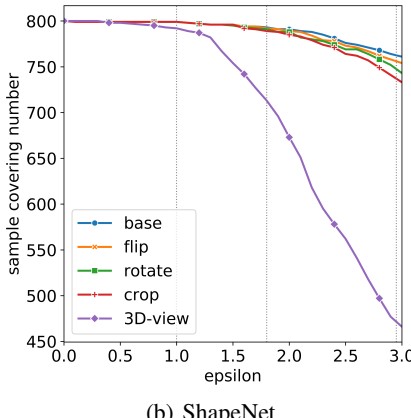

(a) CIFAR-10                            (b) ShapeNet

Figure 2: Estimated sample covering numbers induced by different data transformations at different resolutions $\epsilon$. "base" indicates no transformation. Note that as $\epsilon$ increases, it starts to exceed the $L_2$ distance between some images and thus some images get covered by others without doing any transformation. Three vertical dashed lines indicate the maximum resolution $\epsilon$ at which the "base" yields a certain sample covering number, and from left to right they are $100\%n$, $99\%n$, $95\%n$.

capture the Lipschitz constant change and use it to normalize the sample covering number for better data transformation selections (see Appendix E.5.2).

## 6 Experiments

In this section, we implement the sample cover estimation algorithm and verify the effectiveness of using sample covering numbers to guide the data transformations selection. We first estimate the sample covering number induced by different types of data transformations on common image datasets. Then, we investigate the actual generalization benefit for models invariant to those data transformations and analyze the correlation[4].

**Datasets.** We report experimental results on CIFAR-10 [29] and ShapeNet [10] in this section, and relegate results on CIFAR-100 and Restricted ImageNet to Appendix E.5.1. ShapeNet is a large-scale 3D data repository which enables us to do more complex data transformations (e.g., change of 3D-view) beyond the common 2D geometric transformations. The work [12] provides 24 multi-view pre-rendered images for each 3D object in 10 chosen categories. For convenience, we use those images to approximate the random perturbation of the 3D-view.

**Data transformations.** We evaluate some commonly used data transformations with typical parameter settings which we assume to be label-preserving. We choose *flipping*, *cropping*, and *rotation* on CIFAR-10, and additionally consider the *3D-view* change on ShapeNet. We use the same data transformations with the same parameter settings during estimating the sample covering number and evaluating the generalization benefit. Appendix E provides more details of our experimental settings.

### 6.1 Estimation of Sample Covering Numbers

We implement the algorithm in Section 4 to estimate the sample covering number induced by different data transformations. For efficiency, we randomly sample 1000 training images from CIFAR-10 and randomly sample 800 training images from ShapeNet. Appendix E compares results with different sample sizes. We use the Euclidean norm for defining the sample cover. For continuous data transformations, we do uniform random sampling to approximate the orbit of a data point.

Figure 2 illustrates the estimated sample covering numbers induced by different transformations at different resolution $\epsilon$. As the resolution $\epsilon$ increases, the sample covering number $N(\epsilon, \mathcal{S}, \rho_{\mathcal{G}})$ induced

---

[4]Code is available at https://github.com/bangann/understanding-invariance.

| | $n = 100$ | | $n = 1000$ | | $n = all$ | |
|---|---|---|---|---|---|---|
| Model | acc (%) | gap | acc (%) | gap | acc (%) | gap |
| Base | $41.05 \pm 0.52$ | $58.95 \pm 0.52$ | $68.62 \pm 0.90$ | $31.38 \pm 0.90$ | $85.43 \pm 0.35$ | $14.57 \pm 0.35$ |
| Flip | $44.19 \pm 0.74$ | $55.81 \pm 0.74$ | $75.12 \pm 0.20$ | $24.88 \pm 0.20$ | $89.67 \pm 0.24$ | $10.33 \pm 0.24$ |
| Rotate | $47.02 \pm 0.46$ | $52.93 \pm 0.51$ | $76.07 \pm 0.28$ | $23.92 \pm 0.27$ | $89.91 \pm 0.13$ | $10.05 \pm 0.16$ |
| Crop | $\mathbf{50.47 \pm 0.48}$ | $\mathbf{49.53 \pm 0.48}$ | $\mathbf{81.84 \pm 0.12}$ | $\mathbf{18.15 \pm 0.11}$ | $\mathbf{92.52 \pm 0.08}$ | $\mathbf{7.48 \pm 0.08}$ |

Table 1: Classification accuracy and generalization gap (the difference between training and test accuracy) for ResNet18 on CIFAR-10. The number $n$ denotes the sample size per class.

| | $n = 100$ | | $n = 1000$ | | $n = all$ | |
|---|---|---|---|---|---|---|
| Model | acc (%) | gap | acc (%) | gap | acc (%) | gap |
| Base | $67.75 \pm 2.02$ | $32.25 \pm 2.02$ | $83.33 \pm 0.38$ | $16.67 \pm 0.38$ | $91.81 \pm 0.22$ | $8.18 \pm 0.22$ |
| Flip | $69.75 \pm 1.55$ | $30.25 \pm 1.55$ | $84.24 \pm 0.30$ | $15.76 \pm 0.30$ | $92.07 \pm 0.20$ | $7.92 \pm 0.20$ |
| Rotate | $70.25 \pm 1.19$ | $29.50 \pm 1.15$ | $83.93 \pm 0.38$ | $15.94 \pm 0.35$ | $91.85 \pm 0.20$ | $8.03 \pm 0.26$ |
| Crop | $74.88 \pm 1.03$ | $23.53 \pm 1.30$ | $86.13 \pm 0.39$ | $13.75 \pm 0.32$ | $92.64 \pm 0.12$ | $7.17 \pm 0.19$ |
| 3D-View | $\mathbf{78.13 \pm 1.31}$ | $\mathbf{14.94 \pm 1.76}$ | $\mathbf{88.79 \pm 0.34}$ | $\mathbf{8.38 \pm 0.79}$ | $\mathbf{94.38 \pm 0.08}$ | $\mathbf{3.09 \pm 0.10}$ |

Table 2: Classification accuracy and generalization gap (the difference between training and test accuracy) for ResNet18 on ShapeNet. The number $n$ denotes the sample size per class.

by any data transformation starts to decrease, indicating a smaller-sized sample cover needed to cover the entire dataset. Meanwhile, different transformations behave differently. On CIFAR-10, cropping induces the smallest sample covering number. On ShapeNet, 3D-view transformation induces the smallest sample covering number and the gap is significant. Our propositions suggest that data transformations which induce smaller sample covering numbers tends to bring more generalization benefit for the corresponding invariant models. Therefore, Figure 2 indicates that models should generalize well if it is invariant to 3D-view transformation on ShapeNet or to cropping on CIFAR-10.

## 6.2 Evaluation of Generalization Benefit

We now evaluate the actual generalization performance of invariant models to verify if the sample covering number is a good suitability measurement. We use ResNet18 [25] on both datasets and discuss the influence of model class's implicit bias in Appendix E. A simple method to learn invariant models is to do data augmentation. The augmented loss function is $\mathcal{L}_{aug}(\boldsymbol{x}) = \mathcal{L}(f(g(\boldsymbol{x})))$, where $f(\cdot)$ denotes the model and $g(\boldsymbol{x})$ denotes a randomly sampled example in $\boldsymbol{x}$'s orbit induced by transformation $\mathcal{G}$. We use this method on CIFAR-10 and ShapeNet and show results in Table 1 and 2.

**Sample covering number correlates well with generalization benefit.** We use the generalization gap (the gap between training accuracy and test accuracy) to measure actual generalization benefit. Compared with the baseline, invariant models show an improved reduced generalization gap and also improved test accuracy. On CIFAR-10, cropping-invariant model shows the smallest generalization gap and the highest accuracy. On ShapeNet, the model that is invariant to 3D-view changes shows the smallest generalization gap and the highest accuracy, especially when the training data size is small. By comparing results in Figure 2 and Table 1-2, we observe a clear correlation between the smaller sample covering number and better generalization benefit. This verifies our proposition — invariance to more suitable data transformations gives the model more generalization benefit.

**Model invariance indeed improves after learning.** To verify that the improved generalization is indeed brought by the model invariance, we further enforce the invariance using the invariance regularization loss similar to [48, 51]: $\mathcal{L} = \mathcal{L}_{cls}(f(\boldsymbol{x})) + \lambda \text{KL}(f(\boldsymbol{x}), f(g(\boldsymbol{x})))$. Specifically, in addition to minimizing the classification loss on original images, we penalize the model by minimizing the KL divergence between model outputs on original images and on transformed ones. At test time, we use $\mathcal{L}_{inv}(\boldsymbol{x}) = \mathbb{E}_{g_1, g_2 \in \mathcal{G}}[\text{KL}(f(g_1(\boldsymbol{x})), f(g_2(\boldsymbol{x})))]$ to evaluate the model invariance under transformation $\mathcal{G}$. Table 3 shows that, as we increase the invariance penalty by increasing $\lambda$, invariant models enjoy smaller generalization gap. Moreover, the decreased invariance loss and increased

| $\lambda$ | train acc (%) | test acc (%) | gap | $\mathcal{L}_{inv}$ | $\mathcal{A}_{inv}(\%)$ |
|---|---|---|---|---|---|
| 0 | $99.99 \pm 0.01$ | $91.81 \pm 0.22$ | $8.19 \pm 0.22$ | $0.0548 \pm 0.0028$ | $62.0 \pm 0.6$ |
| 0.01 | $99.98 \pm 0.00$ | $92.77 \pm 0.16$ | $7.21 \pm 0.16$ | $0.0290 \pm 0.0029$ | $74.78 \pm 1.61$ |
| 0.1 | $99.99 \pm 0.00$ | $93.87 \pm 0.19$ | $6.11 \pm 0.19$ | $0.0152 \pm 0.0003$ | $83.12 \pm 0.50$ |
| 0.3 | $99.98 \pm 0.00$ | $94.23 \pm 0.11$ | $5.76 \pm 0.11$ | $0.0121 \pm 0.0003$ | $85.10 \pm 0.20$ |
| 1 | $99.58 \pm 0.04$ | $94.68 \pm 0.09$ | $4.90 \pm 0.09$ | $0.0095 \pm 0.0001$ | $86.94 \pm 0.08$ |
| 3 | $97.74 \pm 0.19$ | $94.48 \pm 0.19$ | $3.26 \pm 0.09$ | $0.0060 \pm 0.0003$ | $88.15 \pm 0.18$ |
| 10 | $95.67 \pm 0.26$ | $93.56 \pm 0.29$ | $2.11 \pm 0.04$ | $0.0037 \pm 0.0002$ | $89.20 \pm 0.16$ |
| 100 | $92.89 \pm 0.25$ | $91.85 \pm 0.26$ | $1.03 \pm 0.03$ | $0.0018 \pm 0.0001$ | $89.82 \pm 0.10$ |

Table 3: Evaluation of ResNet18 on ShapeNet under 3D-view transformations. $\mathcal{L}_{inv}$ denotes the test invariance loss. $\mathcal{A}_{inv}$ denotes the test consistency accuracy (indicating whether model's prediction is unchanged after data transformation) under the worst-case data transformations.

consistency accuracy verify that the model invariance indeed improves after training, supporting that the generalization benefit is brought by the model invariance.

## 7 Related Work

**Understandings from the input space perspective.** One line of work characterizes the input space of invariant models. [3, 4] show that the invariant representations equivalently reduce the input dimension for downstream tasks and thus significantly reduce the model complexity (exponential in input dimensions) of downstream linear models. [42, 39] essentially factorize the input space into the product of a base space and a finite set of data transformations. Since the covering number needed to cover the base space is smaller, the associated generalization bound for invariant models is reduced. Compared with these works, our work tries to cover the sample instead of the input space which circumvents the strong dependence on input dimensions and also enables practical evaluation.

**Understandings from the function space perspective.** Another line of work directly characterizes the function space of invariant models. [30] uses PAC-Bayes to show the reduction of generalization upper bound when the model class is symmetrized to be invariant. [19] analyzes the function space under feature averaging operator and shows the first strict generalization gap (instead of upper bound) via a linear model. This line of work, while being elegant, so far restricts the model invariance to be obtained exclusively via feature averaging.

Note that the categorization of different understanding perspectives are only for presentation convenience and hold no formal distinctions. We also mention some work that studies the model invariance but does not focus on understanding the benefit. [1] proves that the VC-dimension of an invariant model cannot be larger than its counterpart. [9] characterizes the general functional representations of invariant probability distributions as well as neural network structures that implement them. [11] uses group theory to show the benefit of learning with data-augmented loss. In the predicting generalization competition at NeurIPS 2020 [26], the runner-up team [28] shows that model robustness against data transformations can be used as a decent empirical proxy for predicting models' generalization performance. [37] enforce model invariance to learned data transformations that capture inter-domain variation to improve the out-of-distribution generalization. [8] propose to select data transformations automatically from model training via optimizing a parameterized distributions of data transformations. Interestingly, our sample covering number may be used to determine their regularization coefficients for better trade-offs.

## 8 Conclusion

In this paper, we study the generalization benefit of model invariance by deriving model complexity bounds based on the sample cover induced by data transformations. We also propose an algorithm to estimate the sample cover and empirically show that the sample covering number can guide the data transformations selection. Hopefully, this work will fuel the exploration of more suitable data transformations on specific datasets. An interesting direction for future work is to consider the implicit bias of model classes to better understand the generalization benefit of model invariance.

## Acknowledgements

This work is supported by a startup fund from the Department of Computer Science of the University of Maryland, National Science Foundation IIS CRII Award, DOD-DARPA-Defense Advanced Research Projects Agency Guaranteeing AI Robustness against Deception (GARD), Air Force Material Command, and Adobe, Capital One and JP Morgan faculty fellowships.

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
