# A    Complexity Measurements and Generalization Bounds

In this section, we provide additional details on complexity measurements and generalization bounds.

The following lemma bounds the empirical Rademacher complexity of a function class $\mathcal{H}$ via the covering number of $\mathcal{H}$ evaluated at the sample $\mathcal{S}$.

**Lemma A.1** (Dudley's Entropy Integral Theorem [17, 33]). Let $\mathcal{H}$ be a function class from $\mathcal{X}$ to $\mathbb{R}$. Then, for any $\alpha > 0$,

$$\mathfrak{R}_{\mathcal{S}}(\mathcal{H}) \le 4\alpha + 12 \int_{\alpha}^{\infty} \sqrt{\frac{\log N\big(\tau, \mathcal{H}, L_2(\mathbb{P}_{\mathcal{S}})\big)}{n}} d\tau.$$

The following theorem provides a uniform generalization bound for a function class via empirical Rademacher complexity.

**Theorem A.2** ([7, 33]). Let $\mathcal{H}$ be a function class from $\mathcal{X}$ to $[0, B]$. For any $\delta > 0$, with probability at least $1 - \delta$ over the draw of a sample $\mathcal{S}$ with size $n$ according to data distribution $\mathcal{D}$, the following holds for any $h \in \mathcal{H}$:

$$R(h) \le R_{\mathcal{S}}(h) + 2B\mathfrak{R}_{\mathcal{S}}(\mathcal{H}) + 3B\sqrt{\frac{\log \frac{2}{\delta}}{2n}} \tag{A.1}$$

We can plug the refined Rademacher complexity bounds in Proposition B.1 and Theorem C.1 into (A.1) to get refined generalization bounds for certain invariant models.

# B    Proofs

We first prove Theorem 3.4, and then Proposition 3.3.

## B.1    Proof of Theorem 3.4

*Proof of Theorem 3.4.* The general idea of this proof is to show that any cover of *a model class evaluated at a sample cover* also generates a same-sized cover of *the model class evaluated at the original sample* with some additional approximation error. The covering number inequality in (3.4) then follows by taking the minimization over all covers of the model class evaluated at the original sample. Since this proof includes some tedious notations, we first restate the problem setup and then go to the details.

**Problem setup.** Let $\mathcal{S} = \{\boldsymbol{x}_1, \boldsymbol{x}_2, \ldots, \boldsymbol{x}_n\}$ be a sample of size $n$. Let $\widehat{\mathcal{S}} \subseteq \mathcal{S}$ be an $\epsilon$-cover of $\mathcal{S}$ with respect to $\rho_{\mathcal{G}}$ and has size $m$. Without loss of generality, we then vectorize $\mathcal{S}$ and $\widehat{\mathcal{S}}$ for notation simplicity. Denote by $S = (\boldsymbol{x}_1, \boldsymbol{x}_2, ..., \boldsymbol{x}_n)^T$ the vectorized sample associated with $\mathcal{S}$ in some arbitrary but fixed order. Denote by $\widehat{S} = (\widehat{\boldsymbol{x}}_1, \widehat{\boldsymbol{x}}_2, ..., \widehat{\boldsymbol{x}}_m)^T$ the vectorized sample cover associated with $\widehat{\mathcal{S}}$ in some arbitrary but fixed order. $S$ and $\widehat{S}$ thus define a matrix $\mathbf{P}$ below indicating how $\widehat{S}$ approximately recovers $S$:

$$\mathbf{P} = (p_{ij}) \in \mathbb{R}^{n \times m} \quad \text{such that} \quad p_{ij} = \left\{ \begin{array}{ll} 1, & \text{if } \boldsymbol{x}_i \in \mathcal{S} \text{ is approximated by } \widehat{\boldsymbol{x}}_j \in \widehat{\mathcal{S}} \\ 0, & \text{otherwise} \end{array} \right. .$$

We use arbitrary tie-breaking rule when a data point $\boldsymbol{x} \in \mathcal{S}$ can be approximated by multiple $\widehat{\boldsymbol{x}} \in \widehat{\mathcal{S}}$. Without loss of generality, we also assume that there is no "redundant" element in $\widehat{\mathcal{S}}$ which is not used in recovering $\mathcal{S}$, since otherwise it can be removed from $\widehat{\mathcal{S}}$ for a strictly smaller cardinality. Therefore, by definition, $\mathbf{P}$ has linearly independent columns and thus represents injective mapping from $\mathbb{R}^m$ to $\mathbb{R}^n$. We denote by $S'$ the approximately recovered $S$ generated by $\widehat{S}$: $S' = \mathbf{P}\widehat{S}$. The first line in Table 4 shows the relationship among $\widehat{S}$, $S'$, and $S$.

Based on the definition of $\mathbf{P}$, we now give the precise definition of $p(\boldsymbol{x})$ used in defining the (pseudo)metrics in this theorem.

$$p(\widehat{\boldsymbol{x}}_j) = \sum_{i=1}^{n} p_{ij}, \quad \forall j \in [m].$$

$$\widehat{S} \xrightarrow{\text{generates}} S' = P\widehat{S} \xrightarrow{\text{approximates}} S$$

$$\mathcal{T}(\mathcal{H}_{|\widehat{S}}) \xrightarrow[\textbf{step (I)}]{\text{generates}} \mathcal{T}(\mathcal{H}_{|S'}) \xrightarrow[\textbf{step (II)}]{\text{is also}} \mathcal{T}(\mathcal{H}_{|S})$$

Table 4: A diagram of the proof of Theorem 3.4.

| Space | Vector | Vector (in the cover) |
|---|---|---|
| $(\mathbb{R}^m, \rho_m)$ | $h_{|\widehat{S}} \in \mathcal{H}_{|\widehat{S}}$ | $\widehat{h}_{|\widehat{S}} \in \mathcal{T}(\mathcal{H}_{|\widehat{S}})$ |
| $(\mathbb{R}^n, \rho_n)$ | $h_{|S'} \in \mathcal{H}_{|S'}$ | $\widehat{h}_{|S'} \in \mathcal{T}(\mathcal{H}_{|S'})$ |
| $(\mathbb{R}^n, \rho_n)$ | $h_{|S} \in \mathcal{H}_{|S}$ | $\widehat{h}_{|S} \in \mathcal{T}(\mathcal{H}_{|S})$ |

Table 5: Some notations used in the proof of Theorem 3.4.

We proceed to introduce notations for the model class. Instead of considering the model class $\mathcal{H}$ under the metric induced by the function norm $L_2(\mathbb{P}_S)$ (or $L_2(\mathbb{P}_{\widehat{S}})$), we equivalently consider the evaluation of $\mathcal{H}$ at $S$ (or $\widehat{S}$) under the metric $\rho_n$ (or $\rho_m$) in this proof for notation simplicity. We denote the evaluation of $\mathcal{H}$ at $S$ as $\mathcal{H}_{|S} = \{(h(\boldsymbol{x}_1), \ldots, h(\boldsymbol{x}_n))^T : h \in \mathcal{H}\}$, and similarly its evaluation at $\widehat{S}$ as $\mathcal{H}_{|\widehat{S}} = \{(h(\widehat{\boldsymbol{x}}_1), \ldots, h(\widehat{\boldsymbol{x}}_m))^T : h \in \mathcal{H}\}$. We define the metric $\rho_n$ on $\mathbb{R}^n$ as $\rho_n(u, u') = \frac{1}{\sqrt{n}}\|u - u'\|_2$, and the metric $\rho_m$ on $\mathbb{R}^m$ as $\rho_m(v, v') = \frac{1}{\sqrt{n}}\|(\mathbf{P}^T\mathbf{P})^{\frac{1}{2}}(v - v')\|_2$. Therefore, the covering number notation $N(\tau, \mathcal{H}, L_2(\mathbb{P}_S))$ is equivalent to $N(\tau, \mathcal{H}_{|S}, \rho_n)$, and $N(\tau, \mathcal{H}, L_2(\mathbb{P}_{\widehat{S}_{\mathcal{G},\epsilon}}))$ is equivalent to $N(\tau, \mathcal{H}_{|\widehat{S}}, \rho_m)$. Table 5 shows an overview of these notations.

**Summary.** The proof has the following steps. **(I)** Any cover $\mathcal{T}(\mathcal{H}_{|\widehat{S}})$ of a model class evaluated at the sample cover $\widehat{S}$ generates a same-sized cover $\mathcal{T}(\mathcal{H}_{|S'})$ of the model class evaluated at the approximated sample $S'$. **(II)** The cover $\mathcal{T}(\mathcal{H}_{|S'})$ of the model class evaluated at the approximated sample is also a cover $\mathcal{T}(\mathcal{H}_{|S})$ of the model class evaluated at the original sample $S$. **(III)** The covering number inequality follows by taking the minimization over all covers of the model class evaluated at the original sample $S$.

**Step (I).** We first show that any cover $\mathcal{T}(\mathcal{H}_{|\widehat{S}})$ of $\mathcal{H}_{|\widehat{S}}$ generates a set, denoted as $\mathcal{T}(\mathcal{H}_{|S'})$, with the same cardinality. Given any $\mathcal{T}(\mathcal{H}_{|\widehat{S}})$, we construct $\mathcal{T}(\mathcal{H}_{|S'}) = \{P\widehat{h}_{|\widehat{S}} : \widehat{h}_{|\widehat{S}} \in \mathcal{T}(\mathcal{H}_{|\widehat{S}})\}$. Since $\mathbf{P}$ represents injective mapping from $\mathbb{R}^m$ to $\mathbb{R}^n$, we have $|\mathcal{T}(\mathcal{H}_{|S'})| = |\mathcal{T}(\mathcal{H}_{|\widehat{S}})|$ by construction.

Then, we show that $\mathcal{T}(\mathcal{H}_{|S'})$ is a $\tau$-cover of $\mathcal{H}_{|S'}$ with respect to $\rho_n$ if $\mathcal{T}(\mathcal{H}_{|\widehat{S}})$ is a $\tau$-cover of $\mathcal{H}_{|\widehat{S}}$ with respect to $\rho_m$. By the definition of $\mathbf{P}$, it holds that $h_{|S'} = \mathbf{P}h_{|\widehat{S}}$ for any $h \in \mathcal{H}$, and $\mathbf{P}^\dagger\mathbf{P} = \mathbf{I}$ where $\mathbf{P}^\dagger$ is the Moore–Penrose inverse of $\mathbf{P}$ since $\mathbf{P}$ has linearly independent columns. Thus, for any $h_{|S'} \in \mathcal{H}_{|S'}$, we can project it to $\mathcal{H}_{|\widehat{S}}$ by $\mathbf{P}^\dagger h_{|S'}$. Given that $\mathcal{T}(\mathcal{H}_{|\widehat{S}})$ is a $\tau$-cover of $\mathcal{H}_{|\widehat{S}}$ with respect to $\rho_m$, for any $h_{|S'} \in \mathcal{H}_{|S'}$, there exists $\widehat{h}_{|\widehat{S}} \in \mathcal{T}(\mathcal{H}_{|\widehat{S}})$ such that $\rho_m(\mathbf{P}^\dagger h_{|S'}, \widehat{h}_{|\widehat{S}}) \le \tau$. It

follows that

$$\rho_m(\mathbf{P}^\dagger h_{|S'}, \widehat{h}_{|\widehat{S}}) = \frac{1}{\sqrt{n}} \|(\mathbf{P}^T\mathbf{P})^{\frac{1}{2}}(\mathbf{P}^\dagger h_{|S'} - \widehat{h}_{|\widehat{S}})\|_2$$

$$= \frac{1}{\sqrt{n}} \sqrt{(\mathbf{P}^\dagger h_{|S'} - \widehat{h}_{|\widehat{S}})^T(\mathbf{P}^T\mathbf{P})(\mathbf{P}^\dagger h_{|S'} - \widehat{h}_{|\widehat{S}})}$$

$$= \frac{1}{\sqrt{n}} \sqrt{(h_{|S'} - \mathbf{P}\widehat{h}_{|\widehat{S}})^T(h_{|S'} - \mathbf{P}\widehat{h}_{|\widehat{S}})}$$

$$= \frac{1}{\sqrt{n}} \|(h_{|S'} - \mathbf{P}\widehat{h}_{|\widehat{S}})\|_2$$

$$= \rho_n(h_{|S'}, \widehat{h}_{|S'}) \leq \tau,$$

where $\widehat{h}_{|S'} = \mathbf{P}\widehat{h}_{|\widehat{S}}$ is in $\mathcal{T}(\mathcal{H}_{|S'})$ by construction and approximates the given $h_{|S'}$. Therefore, for any $h_{|S'} \in \mathcal{H}_{|S'}$, there exists $\widehat{h}_{|S'} \in \mathcal{T}(\mathcal{H}_{|S'})$ such that $\rho_n(h_{|S'}, \widehat{h}_{|S'}) \leq \tau$, which implies that $\mathcal{T}(\mathcal{H}_{|S'})$ is a $\tau$-cover of $\mathcal{H}_{|S'}$.

**Step (II).** We proceed to show that $\mathcal{T}(\mathcal{H}_{|S'})$ is also a $(\tau + \kappa\epsilon\sqrt{1 - \frac{|\widehat{S}|}{n}})$-cover of $\mathcal{H}_{|S}$. Consider any index $i \in [n]$. Given that $\widehat{S}$ is an $\epsilon$-sample cover of $S$ with respect to $\rho_\mathcal{G}$, we have $\rho_\mathcal{G}(\boldsymbol{x}_i, \boldsymbol{x}_i') = \inf_{\gamma \in \Gamma(\boldsymbol{x}_i, \boldsymbol{x}_i')} \int_\gamma c(r)dr \leq \epsilon$. Moreover, for any $\xi > 0$, by the definition of infimum there exists a path $\gamma_0$ such that $\int_{\gamma_0} c(r)dr \leq \epsilon + \xi$. The following result then shows that the evaluations of any $h \in \mathcal{H}$ at data points $\boldsymbol{x}_i$ and $\boldsymbol{x}_i'$ are close (let $\nabla_{\boldsymbol{x}}h \in \partial h(\boldsymbol{x})$ when $h$ is only subdifferentiable at some $\boldsymbol{x}$):

$$|h(\boldsymbol{x}_i) - h(\boldsymbol{x}_i')| = \int_{\gamma_0} \nabla_{\boldsymbol{x}}h(\boldsymbol{r}) \cdot d\boldsymbol{r}$$

$$\leq \int_{\gamma_0} \|\nabla_{\boldsymbol{x}}h(\boldsymbol{r})\| \, ds \qquad (ds = \|d\boldsymbol{r}\|)$$

$$= \int_{\gamma_0} \|\nabla_{\boldsymbol{x}}h(\boldsymbol{r})\| \, c(\boldsymbol{r})ds \qquad (\text{invariance of } h)$$

$$\leq \kappa \int_{\gamma_0} c(\boldsymbol{r})ds \qquad (\text{Lipschitzness of } h)$$

$$= \kappa(\epsilon + \xi).$$

Since it holds for any $\xi > 0$, we have $|h(\boldsymbol{x}_i) - h(\boldsymbol{x}_i')| \leq \kappa\epsilon$.

Thus, the evaluations of any $h \in \mathcal{H}$ at samples $S$ and $S'$ are close with respect to $\rho_n$:

$$\frac{1}{\sqrt{n}}\|h_{|S} - h_{|S'}\|_2 = \frac{1}{\sqrt{n}} \sqrt{\sum_{i=1}^n (h(\boldsymbol{x}_i) - h(\widehat{\boldsymbol{x}}_i))^2} \leq \frac{1}{\sqrt{n}} \sqrt{(\kappa\epsilon)^2(n - |\widehat{S}|)} = \kappa\epsilon\sqrt{1 - \frac{|\widehat{S}|}{n}}.$$

Therefore, given any $h_{|S} \in \mathcal{H}_{|S}$, we have $h_{|S'} \in \mathcal{H}_{|S'}$ such that $\rho_n(h_{|S}, h_{|S'}) \leq \kappa\epsilon\sqrt{1 - \frac{|\widehat{S}|}{n}}$ and we can find $\widehat{h}_{|S'} \in \mathcal{T}(\mathcal{H}_{|S'})$ such that $\rho_n(h_{|S'}, \widehat{h}_{|S'}) \leq \tau$ since $\mathcal{T}(\mathcal{H}_{|S'})$ is an $\tau$-cover of $\mathcal{H}_{|S'}$. It then follows that $\widehat{h}_{|S'}$ approximates $h_{|S}$:

$$\rho_n(h_{|S}, \widehat{h}_{|S'}) \leq \rho_n(h_{|S}, h_{|S'}) + \rho_n(h_{|S'}, \widehat{h}_{|S'}) \leq \tau + \kappa\epsilon\sqrt{1 - \frac{|\widehat{S}|}{n}},$$

which implies that $\mathcal{T}(\mathcal{H}_{|S'})$ is a $(\tau + \kappa\epsilon\sqrt{1 - \frac{|\widehat{S}|}{n}})$-cover of $\mathcal{H}_{|S}$.

**Step (III).** The final covering number inequality proceeds as follows. Note that any $\tau$-cover $\mathcal{T}(\mathcal{H}_{|\widehat{S}})$ of $\mathcal{H}_{|\widehat{S}}$ generates an $(\tau + \kappa\epsilon\sqrt{1 - \frac{|\widehat{S}|}{n}})$-cover $\mathcal{T}(\mathcal{H}_{|S})$ of $\mathcal{H}_{|S}$ such that $|\mathcal{T}(\mathcal{H}_{|\widehat{S}})| = |\mathcal{T}(\mathcal{H}_{|S})|$. The

set of all covers of $\mathcal{H}_{|\widehat{S}}$ then generates a set of covers of $\mathcal{H}_{|S}$, which further constitutes a subset of all covers of $\mathcal{H}_{|S}$. Thus, we have the following covering number inequality:

$$
\begin{aligned}
& N\left(\tau + \kappa\epsilon\sqrt{1 - \frac{|\widehat{S}|}{n}}, \mathcal{H}_{|S}, \rho_n\right) \\
& = \min\{|\mathcal{T}(\mathcal{H}_{|S})| : \mathcal{T}(\mathcal{H}_{|S}) \text{ is a cover of } \mathcal{H}_{|S}\} \\
& \leq \min\{|\mathcal{T}(\mathcal{H}_{|S})| : \mathcal{T}(\mathcal{H}_{|S}) \text{ is a cover of } \mathcal{H}_{|S} \text{ and is generated by } \mathcal{T}(\mathcal{H}_{|\widehat{S}})\} \\
& = \min\{|\mathcal{T}(\mathcal{H}_{|\widehat{S}})| : \mathcal{T}(\mathcal{H}_{|\widehat{S}}) \text{ is a cover of } \mathcal{H}_{|\widehat{S}}\} \\
& = N(\tau, \mathcal{H}_{|\widehat{S}}, \rho_m).
\end{aligned}
$$

Since this inequality holds for any resolution $\epsilon$ and any $\epsilon$-sample cover, taking the infimum over all resolutions and sample covers and replacing variables then yields the inequality in (3.4).

$\square$

## B.2 Proof of Proposition 3.3

We first provide a lemma for proving Proposition 3.3. Note that theorem 3.4 directly leads to a refined empirical Rademacher complexity bound in terms of the covering number of $\mathcal{H}$ evaluated at the sample cover. The following lemma is a weaker but simpler version. We can set $\epsilon \to 0$ as $n \to \infty$ to further suppress the additional error term on large samples.

**Lemma B.1** (Refined Rademacher complexity of $\mathcal{G}$-invariant $\mathcal{H}$). Let $\mathcal{S} = \{x_i\}_{i=1}^n$ be a sample of size $n$. Let $\mathcal{H}$ be a model class such that every $h \in \mathcal{H}$ is $\kappa$-Lipschitz with respect to $\|\cdot\|$ and invariant to $\mathcal{G}$. Given any $\epsilon > 0$, $\alpha > 0$, let $\widehat{S}_{\mathcal{G},\epsilon}$ be an $\epsilon$-cover of $\mathcal{S}$. Then

$$
\mathfrak{R}_{\mathcal{S}}(\mathcal{H}) \leq 4\kappa\epsilon\sqrt{1 - \frac{|\widehat{S}_{\mathcal{G},\epsilon}|}{n}} + 4\alpha + 12\int_\alpha^\infty \sqrt{\frac{\log N\left(\tau, \mathcal{H}, L_2(\mathbb{P}_{\widehat{S}_{\mathcal{G},\epsilon}})\right)}{n}} d\tau. \tag{B.1}
$$

*Proof.* Given any $\alpha > 0$, let $\alpha' = \alpha + \kappa\epsilon\sqrt{1 - \frac{|\widehat{S}_{\mathcal{G},\epsilon}|}{n}}$ and $\tau' = \tau + \kappa\epsilon\sqrt{1 - \frac{|\widehat{S}_{\mathcal{G},\epsilon}|}{n}}$. Plugging (3.4) into Dudley's entropy integral theorem (Lemma A.1) yields

$$
\begin{aligned}
\mathfrak{R}_{\mathcal{S}}(\mathcal{H}) & \leq 4\alpha' + 12\int_{\alpha'}^\infty \sqrt{\frac{\log N\left(\tau', \mathcal{H}, L_2(\mathbb{P}_{\mathcal{S}})\right)}{n}} d\tau' \\
& \leq 4\alpha' + 12\int_{\alpha'}^\infty \sqrt{\frac{\log N\left(\tau' - \kappa\epsilon\sqrt{1 - \frac{|\widehat{S}_{\mathcal{G},\epsilon}|}{n}}, \mathcal{H}, L_2(\mathbb{P}_{\widehat{S}_{\mathcal{G},\epsilon}})\right)}{n}} d\tau' \\
& = 4\alpha' + 12\int_{\alpha' - \kappa\epsilon\sqrt{1 - \frac{|\widehat{S}_{\mathcal{G},\epsilon}|}{n}}}^\infty \sqrt{\frac{\log N\left(\tau, \mathcal{H}, L_2(\mathbb{P}_{\widehat{S}_{\mathcal{G},\epsilon}})\right)}{n}} d\tau' \\
& = 4\kappa\epsilon\sqrt{1 - \frac{|\widehat{S}_{\mathcal{G},\epsilon}|}{n}} + 4\alpha + 12\int_\alpha^\infty \sqrt{\frac{\log N\left(\tau, \mathcal{H}, L_2(\mathbb{P}_{\widehat{S}_{\mathcal{G},\epsilon}})\right)}{n}} d\tau.
\end{aligned}
$$

$\square$

*Proof of Proposition 3.3.* Let $\mathcal{H}$ be an invariant model class mapping from $\mathcal{X}$ to $[-B, B]$ for some $B > 0$. Let $\widehat{S}_{\mathcal{G},0}$ be a sample cover such that $|\widehat{S}_{\mathcal{G},0}| = N(0, \mathcal{S}, \rho_{\mathcal{G}}) = m$.

We construct a $\tau$-cover of $\mathcal{H}$ with respect to $L_2(\mathbb{P}_{\widehat{S}_{\mathcal{G},0}})$ as follows: for every $x \in \widehat{S}_{\mathcal{G},0}$, we cover the output range of $\mathcal{H}$ at $x$ by a set of points

$$
\mathcal{Y} = \{-B + (2k - 1)\tau\}_{k=1}^{\lceil \frac{B}{\tau} \rceil}.
$$

Let $\widehat{\mathcal{H}} = \{\widehat{h} : \mathrm{dom}(h) = \widehat{\mathcal{S}}_{\mathcal{G},0}, \widehat{h}(\boldsymbol{x}) \in \mathcal{Y}, \forall \boldsymbol{x} \in \widehat{\mathcal{S}}_{\mathcal{G},0}\}$. To see that $\widehat{\mathcal{H}}$ is indeed a $\tau$-cover of $\mathcal{H}$ with respect to $L_2(\mathbb{P}_{\widehat{\mathcal{S}}_{\mathcal{G},0}})$, given any $h \in \mathcal{H}$, we choose $\widehat{h} \in \widehat{\mathcal{H}}$ such that $|h(\boldsymbol{x}) - \widehat{h}(\boldsymbol{x})| \leq \tau$ for every $\boldsymbol{x} \in \widehat{\mathcal{S}}_{\mathcal{G},0}$ and thus

$$\|h - \widehat{h}\|_{L_2(\mathbb{P}_{\widehat{\mathcal{S}}_{\mathcal{G},0}})} = \left( \sum_{\boldsymbol{x} \in \widehat{\mathcal{S}}_{\mathcal{G},0}} \frac{p(\boldsymbol{x})}{n} \left(h(\boldsymbol{x}) - \widehat{h}(\boldsymbol{x})\right)^2 \right)^{\frac{1}{2}}$$

$$\leq \left( \frac{\sum_{\boldsymbol{x} \in \widehat{\mathcal{S}}_{\mathcal{G},0}} p(\boldsymbol{x})}{n} \tau^2 \right)^{\frac{1}{2}}$$

$$\leq \tau$$

Therefore, for $\tau < B$, the covering number of $\mathcal{H}$ satisfy

$$N\left(\tau, \mathcal{H}, L_2(\mathbb{P}_{\widehat{\mathcal{S}}_{\mathcal{G},0}})\right) \leq \lceil \frac{B}{\tau} \rceil^m \leq \left(\frac{B}{\tau} + 1\right)^m \leq \left(\frac{2B}{\tau}\right)^m,$$

whereas for $\tau > B$, we have $N\left(\tau, \mathcal{H}, L_2(\mathbb{P}_{\widehat{\mathcal{S}}_{\mathcal{G},0}})\right) \leq \lceil \frac{B}{\tau} \rceil^m \leq 1$.

Note that Proposition B.1 holds for any model class if we set $\epsilon = 0$. Plugging $N\left(\tau, \mathcal{H}, L_2(\mathbb{P}_{\widehat{\mathcal{S}}_{\mathcal{G},0}})\right)$ into Proposition B.1 and setting $\epsilon = 0, \alpha = 0$, we have

$$\mathfrak{R}_{\mathcal{S}}(\mathcal{H}) \leq 12 \int_0^\infty \sqrt{\frac{\log N\left(\tau, \mathcal{H}, L_2(\mathbb{P}_{\widehat{\mathcal{S}}_{\mathcal{G},0}})\right)}{n}} d\tau$$

$$\leq 12 \int_0^B \sqrt{\frac{m \log\left(\frac{2B}{\tau}\right)}{n}} d\tau$$

$$= 24B \sqrt{\frac{m}{n}} \int_0^{\frac{1}{2}} \sqrt{\log\left(\frac{1}{t}\right)} dt$$

$$\leq 24B \sqrt{\frac{m}{n}} \int_0^1 \sqrt{\log\left(\frac{1}{t}\right)} dt$$

$$= 24B \sqrt{\frac{m}{n}} \cdot \frac{\sqrt{\pi}}{2}$$

$$\leq 24B \sqrt{\frac{m}{n}}$$

$\square$

### B.3 Binary Coding Constructions of Data Transformations in Proposition 3.5

In Proposition 3.5, given $K$ sets of group-structured data transformations $\{\mathcal{G}^{(1)}, \mathcal{G}^{(2)}, ..., \mathcal{G}^{(K)}\}$, we provide a uniform bound for any $h$ in model class and any set of data transformations. Here, we extent it to any set of combinatorial data transformations. Given a pool of $L$ types of group-structured data transformations $\{\mathcal{G}^{(1)}, \mathcal{G}^{(2)}, ..., \mathcal{G}^{(L)}\}$ (e.g., rotation, flipping), we construct the combinatorial data transformations selection $\mathcal{G}_k$ indexed by $k$ as follows: fix an arbitrary order of the power set of $[L]$ and denote the $k$-th element as $\mathcal{I}_k$. For any $k \in [2^L]$, let $\mathcal{G}_k$ be the direct product of the data transformations selected by $\mathcal{I}_k$: $\mathcal{G}_k = \Pi_{i \in \mathcal{I}_k} \mathcal{G}^{(i)}$. Note that $\mathcal{G}_k$ is also group-structured since direct product preserves the group structure. Proposition 3.5 also applies to these combinatorial data transformations $\{\mathcal{G}_k\}_{k=1}^{2^L}$.

## C  Refined Complexity Analysis for Linear Models

This subsection shows a more interpretable generalization benefit of model invariance by considering linear model class and linear data transformations (e.g., rotation). The following theorem provides a refined model complexity result for the invariant Linear model class.

**Theorem C.1** (Refined Rademacher complexity of $\mathbf{A}$-invariant $\mathcal{H}^{\mathsf{Linear}}$). Let $\mathcal{S} = \{\boldsymbol{x}_i\}_{i=1}^n$ be a sample of size $n$. Let $\mathbf{A}$ be the matrix representation of any linear data transformation. Consider the $L_p$-norm-bounded linear model class $\mathcal{H} = \{\boldsymbol{x} \mapsto \langle \boldsymbol{w}, \boldsymbol{x} \rangle : \boldsymbol{w} \in \mathbb{R}^d, \|\boldsymbol{w}\|_p \leq W\}$ for some $p \geq 1$ and constant $W > 0$. Let $\mathcal{H}^{\mathsf{Linear}} = \{h \in \mathcal{H} : h(\boldsymbol{x}) = h(\mathbf{A}\boldsymbol{x}), \forall \boldsymbol{x} \in \mathbb{R}^d\}$ be the subset of $\mathcal{H}$ that is invariant under transformation $\mathbf{A}$. Then

$$\mathfrak{R}_{\mathcal{S}}(\mathcal{H}^{\mathsf{Linear}}) = \frac{W}{n} E_\sigma \left[ \inf_{\boldsymbol{\eta} \in \mathbb{R}^d} \|\boldsymbol{u}_\sigma + (\mathbf{A} - \mathbf{I})\boldsymbol{\eta}\|_q \right], \tag{C.1}$$

where $\boldsymbol{u}_\sigma = \sum_{i=1}^n \sigma_i \boldsymbol{x}_i$ and $\{\sigma_1, \ldots, \sigma_n\}$ are i.i.d. Rademacher random variables.

*Proof.* The linearity of the model class $\mathcal{H}$ allows us to translate the model invariance to an explicit model class constraint and then precisely compute the Rademacher complexity.

To see that the model invariance, $\langle \boldsymbol{w}, \boldsymbol{x} \rangle = \langle \boldsymbol{w}, \mathbf{A}\boldsymbol{x} \rangle$ for all $\boldsymbol{x} \in \mathbb{R}^d$, is equivalent to an explicit model class constraint $\boldsymbol{w} = \mathbf{A}^T \boldsymbol{w}$, we can choose $\boldsymbol{x}$ to be elements in the standard basis of $\mathbb{R}^d$ and conclude that corresponding entries in $\boldsymbol{w}$ and $\mathbf{A}^T \boldsymbol{w}$ are equal.

Then we precisely compute the Rademacher complexity of $\mathcal{H}$. Let $\boldsymbol{u}_\sigma = \sum_{i=1}^n \sigma_i \boldsymbol{x}_i$, we have

$$\begin{aligned}
\mathfrak{R}_S(\mathcal{H}') &= \mathbb{E}_\sigma \left[ \sup_{\substack{\|\boldsymbol{w}\|_p \leq W \\ (\mathbf{A}^T - \mathbf{I})\boldsymbol{w} = \mathbf{0}}} \frac{1}{n} \sum_{i=1}^n \sigma_i \langle \boldsymbol{w}, \boldsymbol{x}_i \rangle \right] \\
&= \frac{1}{n} \mathbb{E}_\sigma \left[ \sup_{\substack{\|\boldsymbol{w}\|_p \leq W \\ (\mathbf{A}^T - \mathbf{I})\boldsymbol{w} = \mathbf{0}}} \langle \boldsymbol{w}, \boldsymbol{u}_\sigma \rangle \right] \\
&= \frac{1}{n} \mathbb{E}_\sigma \left[ \sup_{\|\boldsymbol{w}\|_p \leq W} \inf_{\boldsymbol{\eta} \in \mathbb{R}^d} \langle \boldsymbol{w}, \boldsymbol{u}_\sigma \rangle + \langle \boldsymbol{w}, (\mathbf{A} - \mathbf{I})\boldsymbol{\eta} \rangle \right] \\
&= \frac{1}{n} \mathbb{E}_\sigma \left[ \inf_{\boldsymbol{\eta} \in \mathbb{R}^d} \sup_{\|\boldsymbol{w}\|_p \leq W} \langle \boldsymbol{w}, \boldsymbol{u}_\sigma + (\mathbf{A} - \mathbf{I})\boldsymbol{\eta} \rangle \right] \tag{$\star$} \\
&= \frac{W}{n} E_\sigma \left[ \inf_{\boldsymbol{\eta} \in \mathbb{R}^d} \|\boldsymbol{u}_\sigma + (\mathbf{A} - \mathbf{I})\boldsymbol{\eta}\|_q \right], \tag{Dual norm}
\end{aligned}$$

where the equality in ($\star$) holds by the von Neumann-Fan minimax theorem, since $\{\boldsymbol{\eta} : \boldsymbol{\eta} \in \mathbb{R}^d\}$ is convex, $\{\boldsymbol{w} : \|\boldsymbol{w}\|_p \leq W\}$ is compact and convex, and $\langle \boldsymbol{w}, \boldsymbol{u}_\sigma + (\mathbf{A} - \mathbf{I})\boldsymbol{\eta} \rangle$ is bi-linear in $\boldsymbol{w}$ and $\boldsymbol{\eta}$. $\square$

**Remark C.2.** As a comparison, the Rademacher complexity of the general linear model class $\mathcal{H}$ is $\mathfrak{R}_{\mathcal{S}}(\mathcal{H}) = \frac{W}{n} E_\sigma \left[ \|\boldsymbol{u}_\sigma\|_q \right]$. Note that we always have the model complexity gap $\mathfrak{R}_{\mathcal{S}}(\mathcal{H}) - \mathfrak{R}_{\mathcal{S}}(\mathcal{H}^{\mathsf{Linear}}) \geq 0$ in Theorem C.1 (as one can check by taking $\boldsymbol{\eta} = \mathbf{0}$ in (C.1)) and the gap can also be made strict in many cases.

The following proposition gives a more interpretable result by further considering the $L_2$-norm-bounded linear model class.

**Proposition C.3** (Refined Rademacher complexity of $L_2$-norm-bounded $\mathbf{A}$-invariant $\mathcal{H}^{\mathsf{Linear}}$). Let $\mathcal{H}^{\mathsf{Linear}}$ be the $L_2$-norm-bounded linear model class that is invariant under transformation $\mathbf{A}$ for some constant $W > 0$ (i.e., $p = 2$ in Theorem C.1). Then

$$\mathfrak{R}_{\mathcal{S}}(\mathcal{H}^{\mathsf{Linear}}) = \frac{W}{n} E_\sigma \left[ \|\mathbf{P}\boldsymbol{u}_\sigma\|_2 \right], \tag{C.2}$$

where $\mathbf{P} = \mathbf{I} - (\mathbf{A} - \mathbf{I})(\mathbf{A} - \mathbf{I})^\dagger$ and $(\mathbf{A} - \mathbf{I})^\dagger$ is the Moore–Penrose inverse of $\mathbf{A} - \mathbf{I}$.

*Proof.* Proposition C.3 follows from the least square solution to Theorem C.1 (with $p = 2$). $\square$

**Remark C.4.** Proposition [C.3] shows that the improvement in model complexity (and thus the generalization bound) for linear invariant models depends both on the sample and on data transformations. The matrix $\mathbf{P}$ in ([C.2]) is essentially the orthogonal projection matrix that projects the weighted sum of data $\boldsymbol{u}_\sigma$ onto the null space of $(\mathbf{A} - \mathbf{I})^T$. Intuitively, the linear data transformation $\mathbf{A}$ separates each input $\boldsymbol{x}$ into two orthogonal components: $\mathbf{P}\boldsymbol{x}$ that is $\mathbf{A}$-invariant, and $\boldsymbol{x} - \mathbf{P}\boldsymbol{x}$ that is $\mathbf{A}$-variant. Linear models that are invariant to $\mathbf{A}$ will ignore the $\mathbf{A}$-variant component and only capture the $\mathbf{A}$-invariant component (otherwise they will not be $\mathbf{A}$-invariant). Suppose that the data distribution has zero mean and bounded variance, then the Rademacher complexity of $\mathcal{H}^{\mathsf{Linear}}$ is upper-bounded by the variance of the $\mathbf{A}$-invariant component in $\boldsymbol{x}$. Therefore, if the data transformation captures most of the data variance, the corresponding invariant models will have much smaller model complexity and thus better generalization performance. We give some examples in Example [C.5].

**Example C.5.** Suppose the data $\boldsymbol{x} \in \mathbb{R}^d$ have Gaussian distribution $\mathcal{N}(0, \sigma^2 \mathbf{I})$. Let $\mathcal{H}$ be the $L_2$-norm-bounded linear model class. Then we have the following Rademacher complexity [33] bounds:

(a) $\mathfrak{R}_n(\mathcal{H}) \leq \sqrt{d} \cdot \frac{W\sigma}{\sqrt{n}}$ for the general $\mathcal{H}$;

(b) $\mathfrak{R}_n(\mathcal{H}') \leq \sqrt{\lceil \frac{d}{2} \rceil} \cdot \frac{W\sigma}{2\sqrt{n}}$ for the flipping-invariant $\mathcal{H}' \subseteq \mathcal{H}$;

(c) $\mathfrak{R}_n(\mathcal{H}'') \leq 1 \cdot \frac{W\sigma}{n}$ for the circular-translation-invariant $\mathcal{H}'' \subseteq \mathcal{H}$. The fast convergence rate of $O(\frac{1}{n})$ guarantees small generalization gap.

# D    Empirical Estimation of Sample Covering Numbers

Detailed steps to estimate sample covering numbers are as follows.

**Step 1.** Compute (or approximate) the direct orbit distance between any two examples in $\mathcal{S}$. The direct orbit distance between any two examples $\boldsymbol{x}_i, \boldsymbol{x}_j \in \mathcal{S}$ is

$$d_{\mathcal{G}}(\boldsymbol{x}_i, \boldsymbol{x}_j) = \|\mathcal{G}(\boldsymbol{x}_i) - \mathcal{G}(\boldsymbol{x}_j)\| = \min_{g_1, g_2 \in \mathcal{G}} \|g_1(\boldsymbol{x}_i) - g_2(\boldsymbol{x}_j)\|.$$

**Step 2.** Compute the $\rho_{\mathcal{G}}$ distance between any two examples in $\mathcal{S}$. Given results in step 1, Computing the $\rho_{\mathcal{G}}$ distance between any two examples can be formulated as a shortest path problem on a complete graph, where each node represents an example and the cost of each edge is the direct orbit distance computed in step 1. The shortest path problem is as follows.

$$\rho_{\mathcal{G}}(\boldsymbol{x}_s, \boldsymbol{x}_t) = \min \sum_{(i,j) \in [|S|]} d_{\mathcal{G}}(\boldsymbol{x}_i, \boldsymbol{x}_j) z_{ij}$$

$$\text{s.t.} \sum_{j \in \delta^+(i)} z_{ij} - \sum_{j \in \delta^-(i)} z_{ji} = \begin{cases} 1, & \text{if } i = s \\ -1, & \text{if } i = t \\ 0, & \text{o.w.} \end{cases}, \quad \forall i \in [|S|]$$

$$\sum_{j \in \delta^+(i)} z_{ij} \leq 1, \quad \forall i \in [|S|]$$

$$z_{ij} \in \{0, 1\}, \quad \forall i, j \in [|S|]$$

where $z_{ij}$ is the binary variable indicating whether the path from $\mathcal{G}(\boldsymbol{x}_i)$ to $\mathcal{G}(\boldsymbol{x}_j)$ belongs to the shortest path, and $\delta^+(i)$, $\delta^-(i)$ are the sets of indices of outgoing and incoming nodes. For each pair of examples, this problem can be solved by shortest path algorithms (e.g., Dijkstra's algorithm) in polynomial time (e.g., $O(n^3)$).

**Step 3.** Construct the pairwise distance matrix $\mathbf{D} \leftarrow [\rho_{\mathcal{G}}(\boldsymbol{x}_i, \boldsymbol{x}_j)]_{i,j}$ and approximate the sample covering number. In experiments, we use modified k-medoids [35] clustering method to find the approximation of $N(\epsilon, \mathcal{S}, \rho_{\mathcal{G}})$. Since the k-medoids algorithm requires the number of clusters as an input, we can assign one heuristically or greedy search it as in Algorithm [1].

**Algorithm 1** `Distance2SampleCoveringNum`: sample covering number approximation based on pairwise distances

---

**Input:** distance matrix $\mathbf{D} \in \mathbb{R}^{|\mathcal{S}| \times |\mathcal{S}|}$, resolution $\epsilon$
**Output:** $\widehat{N}(\epsilon, \mathcal{S}, \rho_{\mathcal{G}})$, an empirical estimation of sample covering number $N(\epsilon, \mathcal{S}, \rho_{\mathcal{G}})$
**Algorithm:**
    Set $k = |\mathcal{S}|$
    Set $scn = |\mathcal{S}|$
    **while** $k > 0$ **do**
      $N = k$
      clusters = KMedoids($\mathbf{D}, k$)    # split $\mathcal{S}$ into $k$ clusters according to $\mathbf{D}$
      **for** every cluster **do**
        **for** every point **do**
          **if** $\mathbf{D}$(point, center) $> \epsilon$ **then**
            $N = N + 1$
          **end if**
        **end for**
      **end for**
      $scn = \min\{N, scn\}$
      $k = k - 1$
    **end while**
    **return** $scn$

---

# E   Experimental Details and Extended Experiments

## E.1   Datasets

We perform our empirical analysis on CIFAR-10, ShapeNet in Section 6 and on CIFAR-100 as well as Restricted ImageNet in Appendix E.5.1.

CIFAR-10 dataset [29] consists of 60000 32x32 colour images in 10 classes, with 6000 images per class. There are 50000 training images and 10000 test images. The categories in CIFAR-10 are: *airplane, automobile, bird, cat, deer, dog, frog, horse, ship, truck*.

ShapeNet[5] [10] is a large-scale 3D model repository. In our experiments, we use a subset of it which contains 10 classes and we resize every image to 32x32. There are 30834 training images and 7709 test images. The categories in this dataset are: *sofa, cabinet, chair, display, loudspeaker, lamp, airplane, table, car, watercraft*. 3D-view transformations could be done by 3D object reconstruction methods, e.g., R2N2 [12], or rendering tools, e.g., PyTorch3D[6]. We use pre-rendered images provided by R2N2[7] to approximate the random perturbations of 3D-view.

CIFAR-100 [29] consists of 60000 32x32 colour images in 100 classes, with 600 images per class. There are 500 training images and 100 testing images per class.

Restricted ImageNet [44] is a subset of ImageNet. It has 8 classes, and each of which is made by grouping a subset of existing, semantically similar ImageNet classes into a super-class. All images are preprocessed into a 64x64 resolution.

## E.2   Data Transformations

In this paper, we consider *flipping, cropping, rotation* and *3D-view* as data transformations in Section 6. We apply them respectively on one image from ShapeNet dataset and illustrate the original and transformed images in Figure 3. For flipping, we only consider the horizontal flipping. For cropping, there are two hyper-parameters, the padding number and the cropping size, that determine a random cropping operation. A image is first padded with the last value at the edge, and then randomly cropped to the certain size. For rotation, we only consider rotating an image around its center. There is one hyper-parameter, degree, that determines a rotation operation. For 3D-view transformations,

---

[5]https://shapenet.org/
[6]https://PyTorch3D.org/
[7]http://3d-r2n2.stanford.edu/

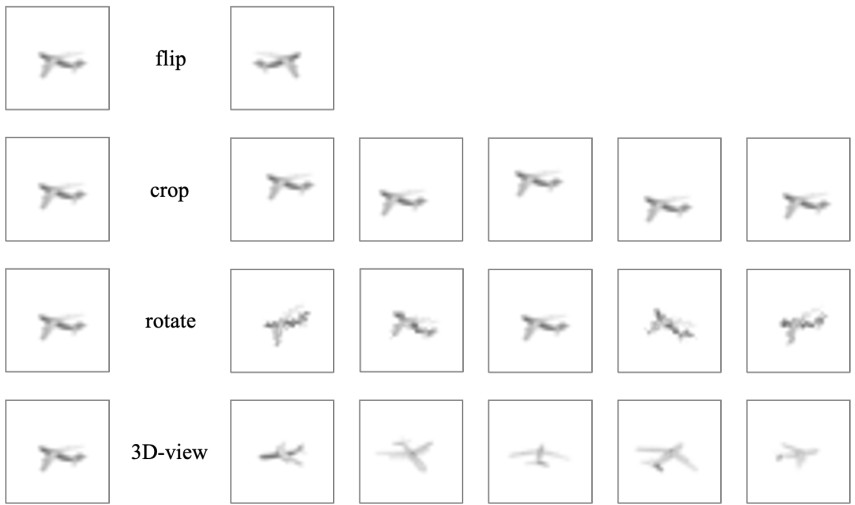

Figure 3: An illustration of data transformations

| Transformation | Hyper-parameters |
| --- | --- |
| Flip | horizontal flip |
| Rotate | degree $\in [-30, 30]$ |
| Crop | padding $= 4$, cropping size $= 32 \times 32$ |
| 3D-view | distance $\in [0.65, 1]$, elevation $\in [25, 30]$, azimuth $\in [0, 360]$ |
| Cutout | value=0.5, scale=0.05, ratio=1 |
| ColorJitter | brightness$\in [0.75, 1.25]$, contrast$\in [0.75, 1.25]$, saturation$\in [0.75, 1.25]$ |

Table 6: Data transformations used in our experiments.

there are three hyper-parameters, distance, elevation and azimuth, that together determine a specific 3D-view. We can interpret the 3D-view as a specific position of the camera which is determined by the distance away from the target point, the elevation angle, and the azimuth angle. As long as the camera's position is determined, we would have the 2D image rendered from that specific viewpoint via R2N2 or PyTorch3D. We also evaluate *cutout* and *color jitter* in Appendix E.5.2. Cutout [15] is a data augmentation method that randomly removes contiguous sections of input images. There are three hyper-parameters that control the size, ratio and pixel values of the rectangle that mask the images. Color jitter is a type of image data transformation where we randomly change the brightness, contrast and saturation of an image which can be controlled by three hyper-parameters.

### E.3 Details on Estimating Sample Covering Numbers

In this paper, we estimate the sample covering numbers induced by different transformations on CIFAR-10, ShapeNet, CIFAR-100 and Restricted ImageNet. Table 6 provides the hyper-parameter settings that we use for data transformations in this paper. These settings are typically used to preserve labels after data transformations in object classification tasks. Continuous data transformations, such as rotation, cropping and 3D-view, contains infinite numbers of elements in the transformation set. To approximate the orbit, we do sampling every 1 degree for rotation and random sampling (50 times) for cropping, cutout and color jitter. We use the set of 24 random multi-view images rendered by R2N2 method to approximate the orbit induced by 3D-view transformations.

### E.4 Details on Evaluating Generalization Benefit

In Section 6.2, we evaluate the generalization benefit of learning model invariance to different data transformations. We consider the object classification task and use ResNet18 model architecture on both datasets. To learn the invariant models, we use two methods: data augmentation and

| | Sample covering number | | | Generalization | |
|---|---|---|---|---|---|
| Model | $\epsilon = 5.7$ | $\epsilon = 7.5$ | $\epsilon = 9.4$ | acc (%) | gap |
| Base | 1000 | 990 | 950 | $60.06 \pm 0.39$ | $39.91 \pm 0.40$ |
| Flip | 1000 | 984 | 945 | $66.49 \pm 0.46$ | $33.48 \pm 0.45$ |
| Rotate | 1000 | 976 | 921 | $67.79 \pm 0.46$ | $32.17 \pm 0.47$ |
| Crop | 995 | 965 | 863 | $72.44 \pm 0.16$ | $27.53 \pm 0.16$ |

Table 7: Sample covering numbers, classification accuracy and generalization gap (the difference between training and test accuracy) for ResNet18 on CIFAR-100.

regularization. In the test phase, we evaluate models on clean test sets without applying any data transformations.

**Data augmentation method.** The taining loss for data augmentation method is $\mathcal{L}_{aug}(\boldsymbol{x}) = \mathcal{L}(f(g(\boldsymbol{x})))$, where $f(\cdot)$ denotes the model and $g(\boldsymbol{x})$ denotes a randomly sampled example in $\boldsymbol{x}$'s orbit induced by transformation $\mathcal{G}$. We use the cross-entropy loss function for $\mathcal{L}$. In each epoch, we randomly sample transformed images as input and preserve ground truth labels. We use SGD optimizer with an initial learning rate of 0.01 and decay the learning rate by 0.1 every 50 epochs. We train each model for 110 epochs and select the best model according to test accuracy. We run independent experiments four times and report the results in Table 1 and 2.

**Regularization method.** The training loss for regularization method is $\mathcal{L}_{reg} = \mathcal{L}_{cls} + \mathcal{L}_{inv} = \mathcal{L}(f(\boldsymbol{x})) + \lambda \text{KL}(f(\boldsymbol{x}), f(g(\boldsymbol{x})))$. Specifically, in addition to minimize the classification loss on the original image, we also regularize the model by minimizing the KL divergence between model's logit outputs on the original image and on the transformed one. The loss function and optimization settings are the same as those in data augmentation method except for the case when $\lambda = 100$. We use learning rate of 0.001 without weight decay and train the model for 500 epoch in that experiment. At test time, we use two metrics to evaluate the model invariance under 3D-view transformations. The first one is the invairance loss, namely $\mathcal{L}_{inv}(\boldsymbol{x}) = \mathbb{E}_{g_1, g_2 \in \mathcal{G}}[\text{KL}(f(g_1(\boldsymbol{x})), f(g_2(\boldsymbol{x})))]$. We approximate the expectation by averaging the KL divergence over the 24 pre-rendered random multi-view images for each original image. The second metric is $\mathcal{A}_{inv}$, namely the consistency accuracy under the worst-case transformation. We have $\mathcal{A}_{inv}(\boldsymbol{x}) = 1$ if model's outputs on data points in $\boldsymbol{x}$'s orbit are consistent, and $\mathcal{A}_{inv}(\boldsymbol{x}) = 0$ otherwise. We also use the 24 pre-rendered multi-view images of $\boldsymbol{x}$ to approximate its orbit. We run independent experiments four times and report the results in Table 3.

## E.5 Extended Experiments

### E.5.1 Experiments on Additional Datasets

To better show the consistency between our theory and practice, we conduct additional experiments on CIFAR-100 [29] and Restricted ImageNet [44]. We randomly sample 1000 examples in training set to evaluate sample covering numbers induced by different data transformations. The settings of data transformations are the same as that in Table 6. We train a ResNet18 with different data augmentations for three times and report results in Table 7 and 8. The results on CIFAR-100 and Restricted ImageNet both support that small sample covering number correlates with small generalization gap.

### E.5.2 Normalization of Sample Covering Numbers

As discussed in Section 5, the proposed sample covering number is a model-agnostic measure that does not consider the potential Lipschitz constant increase induced by data transformations. For example, darken all the images leads to a small sample covering number since the values of all images decrease. However, the Lipschitz constant required for the model is increased to classify closer classes. To mitigate this limitation, we can do normalization for sample covering numbers. Intuitively, the minimum inter-class distance among all class-pairs gives us a clue for the required Lipschitz constant. Therefore, we use the ratio between the minimum inter-class before and after applying data transformations to normalize sample covering numbers. In Table 9, we evaluate 5 types of data transformations including cutout and color jitter. The sample covering number of color jitter is quite

| | Sample covering number | | | Generalization | |
|---|---|---|---|---|---|
| Model | $\epsilon = 14.6$ | $\epsilon = 18.4$ | $\epsilon = 21.6$ | acc (%) | gap |
| Base | 1000 | 990 | 955 | $82.85 \pm 0.42$ | $17.14 \pm 0.42$ |
| Flip | 999 | 986 | 941 | $88.07 \pm 0.39$ | $11.92 \pm 0.39$ |
| Rotate | 998 | 967 | 883 | $88.61 \pm 0.16$ | $11.14 \pm 0.28$ |
| Crop | 995 | 947 | 793 | $91.38 \pm 0.26$ | $8.37 \pm 0.26$ |

Table 8: Sample covering numbers, classification accuracy and generalization gap (the difference between training and test accuracy) for ResNet18 on Restricted ImageNet.

| | SCN | | | Normalized SCN | | | Generalization | |
|---|---|---|---|---|---|---|---|---|
| Model | $\epsilon = 4.9$ | $\epsilon = 6.2$ | $\epsilon = 7.6$ | $\epsilon = 4.9$ | $\epsilon = 6.2$ | $\epsilon = 7.6$ | acc (%) | gap |
| Base | 1000 | 992 | 954 | 1000 | 992 | 954 | $85.43 \pm 0.35$ | $14.57 \pm 0.35$ |
| ColorJitter | 927 | 710 | 372 | 1000 | 994 | 963 | $85.82 \pm 0.33$ | $14.18 \pm 0.33$ |
| Cutout | 999 | 974 | 902 | 1000 | 993 | 963 | $87.24 \pm 0.23$ | $12.75 \pm 0.23$ |
| Flip | 999 | 990 | 946 | 1000 | 995 | 964 | $89.67 \pm 0.24$ | $10.33 \pm 0.24$ |
| Rotate | 999 | 976 | 909 | 1000 | 988 | 939 | $89.91 \pm 0.13$ | $10.05 \pm 0.16$ |
| Crop | 996 | 961 | 863 | 999 | 985 | 909 | $92.52 \pm 0.08$ | $7.48 \pm 0.08$ |

Table 9: Sample covering number (SCN) without and with normalization and generalization performance of ResNet18 on CIFAR-10.

small because it shrinks all the values of images. After normalizing with the minimum inter-class distance, it is larger than that of cropping which align with the actual generalization benefits. This is a heuristic normalization that take potential Lipschitz constant change into consideration. It has limitations such as the normalized sample covering number could exceeds the base one. We leave a better normalization for future work.

### E.5.3 Estimating Sample Covering Numbers with Different Sample Sizes

In Section 6.1, we estimate the sample covering numbers on randomly chosen subsets of the whole training datasets. The sample sizes are 1000 for CIFAR-10 and 800 for ShapeNet. To investigate the impact of sample sizes on estimation, we further estimate the sample covering numbers with different sample sizes on ShapeNet. The results, shown in Figure 4 (a)-(c), show consistent trends and comparisons among different data transformations in all sample size settings. Notably, the 3D-view transformation outperform other type of transformations by a large margin (and indeed yields better generalization benefit as shown in Table 2). Therefore, for guiding the data transformation selection, these results suggest that it suffices to estimate the sample covering number on a small subset of the whole dataset for efficiency.

In addition, Figure 4 (d) shows that the normalized sample covering number decreases as the sample size $n$ increases for fixed $\epsilon$. This result also suggests that we can keep a fixed ratio between the sample covering number and the sample size but gradually shrink the resolution $\epsilon$ as the sample size $n$ grows. For sufficiently large sample size, it is possible to use a very small resolution $\epsilon$ to get a sample covering number that is much smaller than the sample size.

### E.5.4 Influence of Model Class's Implicit Bias on Generalization Benefit

Our proposed sample covering number is a model-agnostic metric to measure the potential generalization benefit of being invariant to certain data transformations. Thus, a natural question is: do all models enjoy the same benefit? To investigate the influence of model class's implicit bias on the generalization benefit, we repeat our experiments using MLP. Different from the ResNet architecture which contains a lot of human priors and engineering work, the 2-layer MLP is among the simplest neural network architectures that better eliminates the influence of architecture's implicit bias. We use the 2-layer MLP which contains 2 hidden layers, each of which has 10000 hidden units. We use ReLU activation for the two hidden layers and do not use common techniques such as batch-normalization

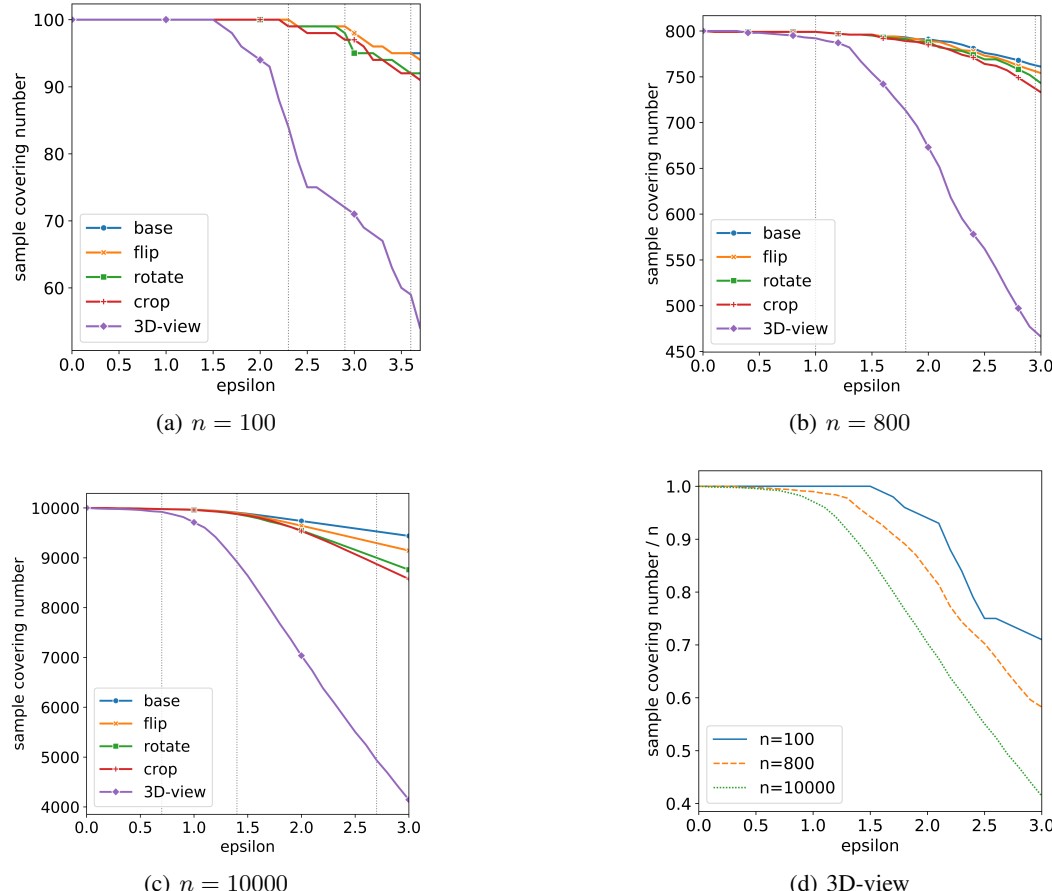

(a) $n = 100$

(b) $n = 800$

(c) $n = 10000$

(d) 3D-view

Figure 4: (a)-(c): Estimated sample covering numbers induced by different data transformations on ShapeNet. $n$ denotes the total sample size. (d): The normalized sample covering number (=sample covering number $/n$) of 3D-view transformations estimated with different sample sizes.

| | $n = 100$ | | $n = 1000$ | | $n = all$ | |
|---|---|---|---|---|---|---|
| Model | acc (%) | gap | acc (%) | gap | acc (%) | gap |
| Base | $64.25 \pm 1.87$ | $20.88 \pm 2.00$ | $77.50 \pm 0.48$ | $21.70 \pm 0.49$ | $86.67 \pm 0.37$ | $12.23 \pm 0.37$ |
| Flip | $65.00 \pm 2.00$ | $13.84 \pm 1.90$ | $78.15 \pm 0.50$ | $16.26 \pm 0.50$ | $87.22 \pm 0.32$ | $9.21 \pm 0.32$ |
| Rotate | $63.50 \pm 2.14$ | $4.88 \pm 2.15$ | $76.70 \pm 0.58$ | $8.98 \pm 0.55$ | $87.00 \pm 0.34$ | $5.12 \pm 0.36$ |
| Crop | $54.56 \pm 1.96$ | $-4.00 \pm 1.80$ | $69.60 \pm 0.42$ | $2.20 \pm 0.42$ | $83.55 \pm 0.32$ | $1.58 \pm 0.36$ |
| 3D-View | $64.75 \pm 1.88$ | $2.25 \pm 1.88$ | $79.20 \pm 0.45$ | $3.18 \pm 0.43$ | $88.28 \pm 0.28$ | $2.00 \pm 0.30$ |

Table 10: Classification accuracy and generalization gap (the difference between training and test accuracy) for MLP on ShapeNet. The number $n$ denotes the sample size per class.

or dropout. We use data augmentation method to train the invariance for the model. The loss function and optimization settings are the same as that used in ResNet18. We run independent experiments four times and report the results in Table 10.

The decreased generalization gaps shown in Table 10 suggest that MLP also benefits from being invariant to data transformations. Moreover, comparisons of the generalization gaps between different transformations are similar to those on ResNet18, indicating the effectiveness and applicability of our proposed metric. Despite the reduced generalization gap, however, MLP trained with invariance suffers from decreased test accuracy in some cases, especially for cropping. This may be due to the limited model capacity of 2-layer MLP learned by SGD. In summary, our proposed sample covering number shows empirical effectiveness in predicting the generalization benefit in a model-agnostic way. Based on our results, we advocate for data transformations that has small sample covering numbers (e.g., 3D-view transformation) and suggest learning the invariance under those data transformations for better generalization performance.