# OpenReview forum: "Understanding the Generalization Benefit of Model Invariance from a Data Perspective"
_NeurIPS.cc/2021/Conference — NeurIPS 2021 Poster_

### Official Review · Reviewer_omim · 2021-06-28

**Rating:** 6
**Confidence:** 3

**Summary:**

The authors tried to understand the generalization benefit based on the invariant transformation by improving the sample covering number. Although the paper focuses on an interesting question (how model invariance helps generalization), it seems that it is still not in its best shape. I believe this paper will benefit from (a) careful polishment on the writing, (b) some more insightful results, and (c) more experiments on some complex datasets.

**Ethical Concerns:**

No Ethical Concerns.

**Limitations And Societal Impact:**

No Societal Impact.

**Main Review:**

I think the paper has the potential to be accepted after some careful polishment. However, I think the current version has some nonnegligible flaws.


## Contributions

1. The authors show that the Rademacher complexity can be reduced when the model is invariant to some transformation (via sample covering number) under Lipschitz and linear regimes.

2. The authors propose an algorithm to estimate the sample covering number empirically.


## Concerns

1. During the proof of Proposition 3.3, since the requirement of Proposition 3.6 is much stronger (a Lipschitz requirement), it cannot be directly applied during the proof of Proposition 3.3.

2. The current conclusion (theory part) in this paper is still not insightful enough. I believe if the transformation can greatly reduce the sample complexity, the generalization could be better. If the authors want to claim "the theory just gives some intuition and the most surprising part is that the practice agreed with the theory", more experiments should be conducted to verify the statements, e.g., CIFAR-100 and ImageNet.

## Writtings.

I believe this paper can benefit from careful writing.

1. For example, is Proposition 3.3 a well-known result? If so, I wonder why the authors give the proof based on Proposition 3.6 (see Line 498-499). Besides, During the proof of Proposition 3.6, I wish the authors can distinguish between Equation 3.3 and Proposition 3.3 (see line 556). When I first read the proofs, I am a little bit puzzled.

2. Line 118-119, why c(r) = |r| instead of |r-x1|?



**Time Spent Reviewing:**

4

---

> ### Author Response · Authors · 2021-08-10
> **Response to Reviewer omim**
>
> We thank Reviewer Omim for the helpful feedback. We address your comments below:
> > **Concern 1:**
> > During the proof of Proposition 3.3, since the requirement of Proposition 3.6 is much stronger (a Lipschitz requirement), it cannot be directly applied during the proof of Proposition 3.3.
>
> **Correct proof but oversimplified writing.** We do not directly apply Proposition 3.6 to the proof of Proposition 3.3, but instead with a minor technical modification as stated in line 498 to avoid duplicated proof. The proof of Proposition 3.3 is correct and we apologize for the confusion caused by our oversimplified and potentially misleading writing. Let us clarify it as below.
>
> By setting $\epsilon=0$, we can generalize Proposition 3.6 to any model class including non-Lipschitz functions as (let $\alpha=0$ for simplicity)
> $$
> \mathfrak{R}\_\mathcal{S}(\mathcal{H}) \le 12\int\_0^\infty \sqrt{\frac{\log N\big(\tau, \mathcal{H}, L_2(\mathbb{P}\_{\hat{\mathcal{S}}\_{\mathcal{G},0}})\big)}{n}} d\tau
> $$
> By plugging the upper bound of $N\big(\tau, \mathcal{H}, L\_2(\mathbb{P}\_{\hat{\mathcal{S}}\_{\mathcal{G},0}})\big)$ derived in line 497 into the inequality above, we can directly get Proposition 3.3.
>
> Note that the proof of Proposition 3.6 is based on the proof of Theorem 3.4, in which only step (II) uses the Lipschitz property. Generalizing Proposition 3.6 to any model class requires a proof that is mostly the same as Proposition 3.6 except for a slightly modified step (II). When $\epsilon=0$, the goal now is to show that $\mathcal{T}(\mathcal{H}\_{|S'})$ is a $\tau$-cover of $\mathcal{H}\_{|S}$  which is much simpler than the previous goal. The proof sketch follows directly by the fact that $h(x\_i)=h(x'\_i)$ for $\epsilon=0$ and that $||h_{|S}-h_{|S'}||_2=0$. In this case, we do not need the Lipschitz property anymore. Thus the result holds for any model class.
>
> We will restate the oversimplified statement in greater detail in the revised version to address your concern. Thank you for this helpful comment.
>
> >**Concern 2:**
> > The current conclusion (theory part) in this paper is still not insightful enough. I believe if the transformation can greatly reduce the sample complexity, the generalization could be better. If the authors want to claim "the theory just gives some intuition and the most surprising part is that the practice agreed with the theory", more experiments should be conducted to verify the statements, e.g., CIFAR-100 and ImageNet.
>
> **Misunderstanding.** We believe that there is a misunderstanding of our contributions. We would argue that - just like other theoretical machine learning studies - a formal justification for the benefit of model invariance is necessary even though it sounds natural and intuitive. The reasons being that formal justification can provide quantitative comparisons between different types of model invariance and also principled ways to handle the potential trade-off between reduced model complexity and increased empirical risk. For example, given a task and a pool of all possible data transformations (including adjustable hyperparameters), how would one choose from those data transformations to train or construct an invariant model to achieve the best generalization? Our work answers this question by the proxy of sample covering numbers and justifies the reason. Our theory is not comprehensive enough at present, but it is the first step towards understanding the benefit of model invariance from the data perspective.
>
>
> **More Experimental Results** The current version of this paper includes experimental results on CIFAR10 and ShapeNet (a 3d dataset). It is a good suggestion to conduct more experiments on other datasets to better show the consistency between our theory and practice. As suggested by the reviewer, we conduct additional experiments on CIFAR100 and Restricted ImageNet [Tsipras et al., 2018] and attach the results below.
>
> * **CIFAR100**
>
> Sample covering numbers (randomly choose 1000 images from the training set):
>
> |Transformation        | $\epsilon$=5.7 | $\epsilon$=7.5 | $\epsilon$=9.4 |
> |--------|:--------------:|:--------------:|:--------------:|
> | Base   | 1000           | 990            | 950            |
> | Flip   | 1000           | 984            | 945            |
> | Rotate | 1000           | 976            | 921            |
> | Crop   | **995**            | **965**            | **863**            |
>
> Classification accuracy and generalization gap for ResNet18 on CIFAR100 (three independent runs):
>
> | Model  | test acc (%) | gap   |
> |--------|---------|-------|
> | Base   | 60.06 $\pm$ 0.39   | 39.91 $\pm$ 0.40 |
> | Flip   | 66.49 $\pm$ 0.46  | 33.48 $\pm$ 0.45 |
> | Rotate | 67.79 $\pm$ 0.46  | 32.17 $\pm$ 0.47 |
> | Crop   | **72.44 $\pm$ 0.16**  | **27.53 $\pm$ 0.16** |
>
> * **Restricted ImageNet**
>
> Sample covering numbers (randomly choose 1000 images from the training set):
>
> |Transformation        | $\epsilon$=14.6 | $\epsilon$=18.4 | $\epsilon$=21.6 |
> |--------|--------------|--------------|--------------|
> | Base   | 1000         | 990          | 955          |
> | Flip   | 999          | 986          | 941          |
> | Rotate | 998          | 967          | 883          |
> | Crop   | **995**          | **947**          | **793**          |
>
> Classification accuracy and generalization gap for ResNet50 on Restricted ImageNet (three independent runs):
>
> | Model  | test acc (%) | gap   |
> |--------|---------|-------|
> | Base   | 82.85 $\pm$ 0.42   | 17.14 $\pm$ 0.42 |
> | Flip   | 88.07 $\pm$ 0.39  | 11.92 $\pm$ 0.39 |
> | Rotate | 88.61 $\pm$ 0.16 | 11.14 $\pm$ 0.28 |
> | Crop   | **91.38 $\pm$ 0.26**  | **8.37 $\pm$ 0.26** |
>
> All the results suggest that the invariance to more suitable data transformations (that have smaller sample covering numbers) gives the model more generalization benefit. We will add these additional results and the detailed settings of experiments to our revised paper.
>
> > **Writings 1.1:**
> > For example, is Proposition 3.3 a well-known result? If so, I wonder why the authors give the proof based on Proposition 3.6 (see Line 498-499).
>
> The Proposition 3.3 is not well-known since it is based on our newly introduced sample covering numbers.
>
> > **Writings 1.2:**
> > Besides, During the proof of Proposition 3.6, I wish the authors can distinguish between Equation 3.3 and Proposition 3.3 (see line 556). When I first read the proofs, I am a little bit puzzled.
>
> We follow a convention (e.g., see [Mohri et al., 2018] and [Chen et al., 2020]) that equations are refered to by bracketed numbers with typically omitted prefix while other objects, including Propositions, are refered to by numbers without brackets and with prefix. We apologize for the conflict of conventions and will distinguish between the two to avoid confusion.
>
> > **Writings 2:**
> > Line 118-119, why c\(r\) = |r| instead of |r-x1|?
>
> The pseudometric $\rho_\mathcal{G}(x_1, x_2)$ finds the path with the shortest (weighted) length. The variable $r$ is the integral variable in the line integral and thus the $\|r\|$ should be corrected as $1$ (not $|r-x_1|$). Thank you for pointing this out.
>
> References:
> 1. Chen, Shuxiao, Edgar Dobriban, and Jane Lee. A Group-Theoretic Framework for Data Augmentation. In Advances in Neural Information Processing Systems. 2020.
> 2. Mohri, Mehryar, Afshin Rostamizadeh, and Ameet Talwalkar. Foundations of Machine Learning. The MIT Press, 2012.
> 3. Tsipras, Dimitris, Shibani Santurkar, Logan Engstrom, Alexander Turner, and Aleksander Madry. Robustness May Be at Odds with Accuracy.” In International Conference on Learning Representations. 2019.

---

> > ### Comment · Reviewer_omim · 2021-08-24
> > **Thanks for the response.**
> >
> > Thanks for your comments which have clarified most of my concerns.
> > I have raised my score to 6 and tend to accept the paper. I am willing to see the next version.

---

### Official Review · Reviewer_Q155 · 2021-07-15

**Rating:** 8
**Confidence:** 3

**Summary:**

The paper provides a bound on the Rademacher complexity of a class of Lipschitz and transformation-invariant functions by upper bounding the covering number of the function class by a similar covering number which uses a "sample cover induced by data transformations". This is the induced covering number which accounts for transformations of the original sample set and can be smaller due to the transformation invariance of the function. This directly gives improved generalization bounds for transformation-invariant functions. The paper gives some helpful intuition and simple cases where these bounds may be applied.

Furthermore, the paper presents experiments in which the sample covering number induced by different transformations is approximated, and the generalization ability of a model which is trained to be invariant to those respective transformations is shown to be strongly correlated with the sample covering number.

**Limitations And Societal Impact:**

Sure

**Main Review:**

I think this is a great paper, with nice theoretical results and simple yet surprisingly insightful experimental results (though I note that only briefly skimmed the proofs). I particularly appreciate that this paper gives not only meaningful improvements to generalization bounds, but also a concrete prescription for how to choose data augmentations to improve generalization in practice.

I do think it's a bit limiting that these results hold only for functions which are *fully* invariant, as this rarely holds without some sort of strong structural prior. Would the proof techniques not work for functions which are *close to* invariant? For example, suppose we assume the function class is $\kappa$-Lipschitz generally, but $\kappa'$-Lipschitz over group transformations, for $\kappa' \ll \kappa$ (or some other measure of "approximate invariance", as Lipschitzness may not appropriately capture such invariance for certain transformations). Would this not allow for a similar result, with an additional error term?

minor nits:
* In line 95, the definition of covering number for function classes is presented with an argument ordering which is permuted; this should be made consistent to match the orderings used later on.
* Line 206, these are classes of functions, so this should be "subset of" ($\subset$), not "element of" ($\in$).

**Time Spent Reviewing:**

2.5

---

> ### Author Response · Authors · 2021-08-10
> **Response to Reviewer Q155**
>
> We thank Reviewer Q155 for the thoughtful feedback and great questions on the applicability of our framework.
>
> > **Q1.** I do think it's a bit limiting that these results hold only for functions which are fully invariant, as this rarely holds without some sort of strong structural prior.
>
> **Extending the applicability.** In addition to constructing fully invariant models using structural prior, for data transformations with group structures, we can also fit any (invariant or non-invariant) models under our framework by compositing them with the invariant loss functions rather than the original loss function $l$. For example, any models composited with the adversarial (worst-case) loss function $\tilde{l}(h(x), y)=\max_{x'\in\mathcal{O}(x)} l(h(x'), y)$ are invariant under the input data transformation and thus enjoy the reduced model complexity.
>
> **Controling the trade-off.** On the other hand, models composited with the adversarial loss are likely to suffer from increased empirical risk due to the adversarially transformed input data. This is the case when the intrinsically invariant models and the approximately invariant models stand out. If the models are intrinsically invariant, then compositing with an worst-case loss will not increase the empirical risk, and the reduced model complexity comes for free. If the models are only approximately invariant in the sense that they are invariant only at some data points, then the empirical risk may increase mildly and the overall generalization would still be improved. This also motivates the invariance training of models even though they do not have built-in structural invariance.
>
> We thank the reviewer for raising this point and will add the applicability discussion in the revised version of this paper.
>
>
> > **Q2.** Would the proof techniques not work for functions which are close to invariant? For example, suppose we assume the function class is κ-Lipschitz generally, but κ′-Lipschitz over group transformations, for κ′≪κ (or some other measure of "approximate invariance", as Lipschitzness may not appropriately capture such invariance for certain transformations). Would this not allow for a similar result, with an additional error term?
>
> **The proof techniques work for the mentioned definition.** The proof techniques directly adapt to the mentioned $\kappa'$-Lipschitz definition of approximate invariance. In fact, we do not even need to modify the proof if we properly change the induced pseudometric $\rho_\mathcal{G}(x_1, x_2)$. To this end, we can set the "transporting cost" when the infinitesimal path segment is inside the orbit from $0$ to $\kappa'/\kappa$. Note that consequently, the estimation algorithm of sample covering number has to add an additional penalty term when searching for the shortest path. This potentially increases the sample covering number and thus reduces the generalization benefit for approximate invariant models compared with fully invariant models.
>
>
> **Different definitions in practice.** The mentioned $\kappa'$-Lipschitz  definition provides a way to extend our framework to approximately invariant models. To further upper-bounding the neural network's Lipschitz constant in practice, we may refer to existing methods such as spectral norm multiplication [Bartlett et al., 2017] and summation [Wei and Ma, 2019]. In addition to the mentioned definition, some related work also provides other definitions of approximate invariance such as “insensitivity” [van der Wilk et al., 2018] and distributional distance [Chen et al., 2020]. That said, we currently do not have a comprehensive description of how those extensions work under our framework and in practice. We suggest using some invariant losses to work around the approximate invariant case as responded in Q1 and leave for future work the other ways. We thank the reviewer for this great question.
>
>
> > minor nits
>
> We have fixed the two typos. Thank you for bringing them up.
>
> References:
> 1. Bartlett, Peter L, Dylan J Foster, and Matus J Telgarsky. Spectrally-Normalized Margin Bounds for Neural Networks. In Advances in Neural Information Processing Systems 30. 2017.
> 2. Chen, Shuxiao, Edgar Dobriban, and Jane Lee. A Group-Theoretic Framework for Data Augmentation. In Advances in Neural Information Processing Systems. 2020.
> 3. van der Wilk, Mark, Matthias Bauer, ST John, and James Hensman. Learning Invariances Using the Marginal Likelihood. In Advances in Neural Information Processing Systems. 2018.
> 4. Wei, Colin, and Tengyu Ma. Data-Dependent Sample Complexity of Deep Neural Networks via Lipschitz Augmentation. In Advances in Neural Information Processing Systems. 2019.

---

### Official Review · Reviewer_TxB5 · 2021-07-20

**Rating:** 6
**Confidence:** 4

**Summary:**

This submission derives model complexity bounds using the idea of transformation-induced sample covering numbers, proposes an algorithm for estimating them, and implements the algorithm on a few data sets.

**Limitations And Societal Impact:**

The authors haven't made clear the extent to which their results focus only on model complexity, nor the potential limitations of their proposed algorithm.

**Main Review:**

Overall, the paper is clearly written, and of relatively high quality and originality. I think its significance could be improved by devoting more space to the algorithm for estimating the sample cover number (see below); by making more explicit the role of invariance in the theoretical results; and fully considering the trade-off between model complexity and empirical risk in generalization bounds.

The theory is missing the connection between invariance-induced model complexity reduction and improved generalization. In particular, both in the main text and in the appendix, the role of invariance isn't clear. Generating the sample cover from a set of transformations is intuitive, but once we have that, the role of invariance doesn't seem particularly important. I'm guessing that I missed something so it would be helpful for the authors to clarify.

Further to that point, all of the theoretical results bound the covering number or Rademacher complexity of $\mathcal{H}$, and of course that's only one part of a generalization bound. One could map all points to a singleton and have a covering number of 1, but that won't do well empirically. What assumptions are needed about the underlying data distribution in order to either i) guarantee that the empirical risk won't increase (at least in probability); or ii) control the trade-off between empirical risk increase and model complexity decrease?

The theoretical development seems correct, and is a nice complementary perspective to the function-space perspective (though I would argue that the distinction isn't meaningful---the function space simplifies because the data space simplifies and vice versa). I am much more enthusiastic about the proposed techniques for estimating sample cover numbers. These are potentially very useful. However, the algorithmic challenges are swept to the appendix, where they are mostly swept under the rug. The challenges should be brought front-and-center, and identified as important for subsequent work. With that in mind, I think the paper would be improved by moving most or all of the linear model class special case to the appendix and devoting additional page space to algorithmic considerations and challenges.

Finally, there are some minor typos scattered throughout the text that the authors should fix.

**Time Spent Reviewing:**

2.5

---

> ### Author Response · Authors · 2021-08-10
> **Response to Reviewer TxB5**
>
> We thank reviewer TxB5 for the insightful feedback and constructive suggestions on improvement.
>
> > **Q1. Fully considering the trade-off between model complexity and empirical risk in generalization bounds**. Further to that point, all of the theoretical results bound the covering number or Rademacher complexity of H, and of course that's only one part of a generalization bound. One could map all points to a singleton and have a covering number of 1, but that won't do well empirically. What assumptions are needed about the underlying data distribution in order to either i) guarantee that the empirical risk won't increase (at least in probability); or ii) control the trade-off between empirical risk increase and model complexity decrease?
>
> Considering the trade-off between model complexity and empirical risk in greater detail is indeed a great point and a classic question. We thank the reviewer for bringing this up and would like to elaborate below.
>
> * **Analysis of the trade-off.** The standard generalization bound, provided in Theorem A.2 and partially restated below, indicates the potential trade-off between model complexity reduction and empirical risk increase.
> $$
> R(h) \le \underbrace{ R\_S(h) }\_{\text{empirical risk}} + \underbrace{ 2\mathfrak{R}\_{S}(\mathcal{H}\_{inv}) }\_{\text{invariant model complexity}} + 3\sqrt{\frac{\log \frac{2}{\delta}}{2m}}
> $$
> As mentioned by the reviewer and briefly addressed in Section 4.1, the invariant model class can have smaller model complexity but we also need to keep the empirical risk low so as to improve the generalization. Intuitively, choosing an inappropriate invariant model class for a given data distribution may hurt the underlying optimal solution and thus increase the empirical risk. One sufficient condition to guarantee that the empirical risk will not increase (e.g., see [Chen et al., 2020]), when considering any measurable function $h:\mathcal{X}\times\mathcal{Y}\rightarrow[0,1]$ and when data transformations have group structure with Haar measure, is that the data distribution is invariant under the transformation: $\mathbb{P}(g(X),Y)=\mathbb{P}(X,Y), \forall g\in\mathcal{G}$. However, a more general condition to characterize the empirical risk requires consideration of specific model class and data properties and thus is beyond the scope of this paper. In our experiments on typical neural networks, simply using the domain knowledge to ensure that the transformation is label preserving consistently maintains a low empirical risk. Nonetheless, guaranteeing a low empirical risk is difficult, so we alternatively discuss how to optimally control the trade-off below.
>
>
> * **Controlling the trade-off.** We can use two principled methods, *structural risk minimization* and *cross-validation*, to control the trade-off between empirical risk increase and model complexity decrease. Suppose we have a set of $t$ different types of data transformations and correspondingly a set of $K=2^t$ different invariant model classes. Our goal is to select the best invariant model class that achieves the optimal trade-off after seeing the dataset. Note that we need a uniform bound that holds for all model class selections so that we can optimally choose the model class after seeing the data. Suppose we index the $K$ invariant model classes in a countable way (e.g., via binary coding for the $t$ different types of data transformations). We may further do the indexing such that small $k$ indicates a simpler model class, though it is not necessary since the term $\sqrt{{\log k}/{m}}$ below is only added to prove convergence. The following result bounds the generalization error uniformly over all model class selections $k\in[K]$.
> $$
> R(h) \le \underbrace{ R\_S(h) + 4\mathfrak{R}\_{S}(\mathcal{H}\_k) + \sqrt{\frac{\log k}{m}} }\_{\text{training objective of structural risk minimization}} + 3\sqrt{\frac{\log \frac{4}{\delta}}{2m}}
> $$
> Using the marked training objective of structural risk minimization to optimize over $k\in[K]$ and $h\in \mathcal{H}\_k$, standard result (e.g., see [Mohri et al., 2018]) show that the solution achieves almost the optimal trade-off between empirical risk increase and invariant model complexity decrease (up to a negligible term $\sqrt{{\log k^*}/{m}}$ for the optimal model class $\mathcal{H}\_{k^*}$). We can also achieve a favorable trade-off using cross-validation which we omit here.
>
> Though this paper mainly focuses on refining the sample-dependent model complexity bound for invariant models, we agree with the reviewer that fully considering the trade-off helps to understand and also highlights the role of invariance. We will add the discussion to the revised version of this paper.
>
> > **Q2. Making more explicit the role of invariance in the theoretical results**. The theory is missing the connection between invariance-induced model complexity reduction and improved generalization. In particular, both in the main text and in the appendix, the role of invariance isn't clear. Generating the sample cover from a set of transformations is intuitive, but once we have that, the role of invariance doesn't seem particularly important. I'm guessing that I missed something so it would be helpful for the authors to clarify.
>
> The model invariance bridges the gap between specific properties of data and the generalization benefit enjoyed by models that are tailored to the data. As shown in Proposition 3.3, the more suitable model invariance, in the sense that the corresponding data transformations yield a small sample covering number, better tightens the model complexity bound. Our response to Q1 also highlights the role of invariance, that is, suitable model invariance reduces the model complexity while keeping the empirical risk low. We note that "good" generalization bounds tend to be heavily data-dependent (e.g., non-margin bound vs. margin bound vs. all-layer margin bound [Wei and Ma, 2020]). Our generalization bound utilizing the model invariance on data thus has the potential to give much tighter results.
>
> More interpretable analysis of the role of invariance requires specific consideration of model class and data distribution - this is the reason why we analyze the linear model class for more interpretable model complexity reduction results. We will add the response in Q1 to the revised version to highlight the role of invariance, while more analyses for other specific model classes are left for future work.
>
> > **Q3. Devoting more space to the algorithm for estimating the sample cover number.**
> I am much more enthusiastic about the proposed techniques for estimating sample cover numbers. These are potentially very useful. However, the algorithmic challenges are swept to the appendix, where they are mostly swept under the rug. The challenges should be brought front-and-center, and identified as important for subsequent work. With that in mind, I think the paper would be improved by moving most or all of the linear model class special case to the appendix and devoting additional page space to algorithmic considerations and challenges.
>
> We agree that the newly proposed algorithm for estimating the sample covering number has its unique value not only for this work but also for subsequent work. It bridges the gap between theory and practice and thus should be highlighted. We will bring the detailed analyses (including challenges and empirical analysis) from appendix to the main text. Thank you for raising this point and also for the great suggestions on the structure of this paper. We will reorganize the paper accordingly in the revised version.
>
>
> > Finally, there are some minor typos scattered throughout the text that the authors should fix.
>
> We have proofread the paper and done our best to fix the typos.
>
>
>
> References:
> 1. Chen, Shuxiao, Edgar Dobriban, and Jane Lee. A Group-Theoretic Framework for Data Augmentation. In Advances in Neural Information Processing Systems. 2020.
> 2. Mohri, Mehryar, Afshin Rostamizadeh, and Ameet Talwalkar. Foundations of Machine Learning. The MIT Press, 2012.
> 3. Wei, Colin, and Tengyu Ma. Improved Sample Complexities for Deep Neural Networks and Robust Classification via an All-Layer Margin. In International Conference on Learning Representations. 2020.

---

> > ### Comment · Reviewer_TxB5 · 2021-08-26
> > **Thanks**
> >
> > Thank you for the thoughtful response.
> >
> > I am optimistic that a future iteration of this paper will be accepted for publication. Without considering the effects on empirical risk more fully, the story is incomplete.

---

### Decision · Program_Chairs · 2021-09-27

**Decision:**

Accept (Poster)

**Comment:**

The reviewers have come to a consensus in favor of this paper being accepted. I agree with this consensus. While there are still some issues described in the reviewers' comments, I expect that these can be addressed satisfactorily in the camera ready, as described in the author response.